# Affimer proteins are versatile and renewable affinity reagents

Christian Tiede[1,2], Robert Bedford[1,2], Sophie J Heseltine[1,2], Gina Smith[1], Imeshi Wijetunga[3], Rebecca Ross[3], Danah AlQallaf[1], Ashley PE Roberts[4], Alexander Balls[1], Alistair Curd[1], Ruth E Hughes[1], Heather Martin[1], Sarah R Needham[5], Laura C Zanetti-Domingues[5], Yashar Sadigh[4], Thomas P Peacock[4], Anna A Tang[1], Naomi Gibson[1], Hannah Kyle[6], Geoffrey W Platt[6], Nicola Ingram[3], Thomas Taylor[1], Louise P Coletta[3], Iain Manfield[1,2], Margaret Knowles[3], Sandra Bell[7], Filomena Esteves[3], Azhar Maqbool[8], Raj K Prasad[8], Mark Drinkhill[8], Robin S Bon[8], Vikesh Patel[9], Sarah A Goodchild[9], Marisa Martin-Fernandez[5], Ray J Owens[10], Joanne E Nettleship[10], Michael E Webb[11], Michael Harrison[12], Jonathan D Lippiat[12], Sreenivasan Ponnambalam[1,2], Michelle Peckham[1,2], Alastair Smith[6], Paul Ko Ferrigno[6], Matt Johnson[6], Michael J McPherson[1,2]*, Darren Charles Tomlinson[1,2]*

[1]School of Molecular and Cellular Biology, University of Leeds, Leeds, United Kingdom; [2]Astbury Centre for Structural and Molecular Biology, University of Leeds, Leeds, United Kingdom; [3]Leeds Institute of Cancer Studies and Pathology, University of Leeds, Leeds, United Kingdom; [4]The Pirbright Institute, Woking, United Kingdom; [5]Central Laser Facility, Research Complex at Harwell, STFC Rutherford Appleton Laboratory, Didcot, United Kingdom; [6]Avacta Life Sciences, Wetherby, United Kingdom; [7]Leeds Institute of Biomedical and Clinical Sciences, University of Leeds, Leeds, United Kingdom; [8]Leeds Institute of Cardiovascular and Metabolic Medicine, University of Leeds, Leeds, United Kingdom; [9]DSTL Porton Down, Salisbury, United Kingdom; [10]Oxford Protein Production Facility UK, Research Complex at Harwell, STFC Rutherford Appleton Laboratory, Didcot, United Kingdom; [11]School of Chemistry, University of Leeds, Leeds, United Kingdom; [12]School of Biomedical Sciences, University of Leeds, Leeds, United Kingdom

*For correspondence: M.J. McPherson@leeds.ac.uk (MJM); d.c.tomlinson@leeds.ac.uk (DCT)

**Abstract** Molecular recognition reagents are key tools for understanding biological processes and are used universally by scientists to study protein expression, localisation and interactions. Antibodies remain the most widely used of such reagents and many show excellent performance, although some are poorly characterised or have stability or batch variability issues, supporting the use of alternative binding proteins as complementary reagents for many applications. Here we report on the use of Affimer proteins as research reagents. We selected 12 diverse molecular targets for Affimer selection to exemplify their use in common molecular and cellular applications including the (a) selection against various target molecules; (b) modulation of protein function in vitro and in vivo; (c) labelling of tumour antigens in mouse models; and (d) use in affinity fluorescence and super-resolution microscopy. This work shows that Affimer proteins, as is the case for other alternative binding scaffolds, represent complementary affinity reagents to antibodies for various molecular and cell biology applications.

**eLife digest** Many of the molecules that are essential for life are too small to be visible inside cells. So, scientists use large complex proteins called antibodies that bind to these molecules to detect whether they are present and show where they are in a cell. As well as being useful tools in experiments, these antibodies can be used to help identify and treat diseases.

The body produces antibodies in response to an infection. The antibodies used in experiments are purified from animal blood, but this method of producing antibodies has flaws. For example, it can be difficult to make identical batches of antibody that always behave in the same way. So scientists have developed "alternative binding proteins" that can be made in the laboratory. These proteins are much less complicated and can be developed more quickly than antibodies, and can easily be adapted for a variety of uses.

An alternative binding protein called an Affimer behaves in a similar way to an antibody by binding tightly to its target molecule, but is much more stable to acidity and high temperature. Tiede et al. have now tested how well the Affimer works in a wide range of different experiments that normally use antibodies to analyse the amount of a particular molecule inside a cell.

The results of the tests show that the Affimer behaves in the same way as antibodies, and sometimes works more effectively. Tiede et al. show that an Affimer can help to reveal how a particular molecule works within a cell, to create detailed pictures of molecules in cells and tissues, and to identify a tumour. It can also be used alongside a new technique called 'super-resolution microscopy' that allows researchers to watch the activity of individual molecules.

Future challenges are to test the Affimer in even more applications and to encourage its wider use by researchers, alongside other alternative binding proteins, as as replacements for some antibodies. This could ultimately lead to the development of faster and more efficient diagnostic, imaging and therapeutic tests.

## Introduction

Our understanding of biological processes at the cellular level has been underpinned by the traditional disciplines of genetics, biochemistry, and molecular biology. Over the last decade, focus has shifted towards large-scale studies of genomes and transcriptomes, the latter as surrogates for cellular proteomes. These combined with high-throughput protein interaction studies, have led to the new discipline of Systems Biology, where proteins are considered in the context of networks of biochemical and developmental pathways. In the network view of protein behaviour, each protein or protein isoform may participate in many protein-protein interactions but available tools that allow researchers to test hypotheses in the biological context are lacking. Technologies such as RNAi and CRISPR-Cas9 that lower or ablate protein expression are important tools, but may cloud the interpretation of a proposed relationship between a given gene product or protein domain and the observed cellular phenotype. The next generation of tools should have the ability to block protein-protein interactions systematically without affecting expression levels.

Commonly used tools for studying protein expression and function include antibodies. Antibodies have proved to be exquisite tools in many applications but there are growing concerns about the difficulty in sourcing validated and renewable antibodies (*Bordeaux et al., 2010*; *Bradbury and Plückthun, 2015*; *Taussig et al., 2007*). While there are over 500,000 different antibodies on the market, it has been reported that up to 75% have either not been validated, show a low level of validation or simply do not perform adequately in certain applications (*Berglund et al., 2008*). In addition, the use of antibodies to block protein function inside living cells is commonly performed, but it is limited owing to the reducing environment of the cells (*Marschall et al., 2015*). Even though antibody fragments, termed intrabodies (*Marschall et al., 2015*) or chromobodies (*Rothbauer et al., 2006*) can be expressed in the cytoplasm of mammalian cells, only a fraction of the repertoire of IgGs are correctly folded in the reducing environment of the cytoplasm (*Biocca et al., 1995*; *Wörn and Plückthun, 2001*), decreasing their efficacy in functional applications (*Marschall et al., 2015*).

Various consortia (*Stoevesandt and Taussig, 2012*) have been established to address the generation and validation of antibodies and their derivatives (*Berglund et al., 2008*; *Renewable Protein Binder Working Group et al., 2011*; *Nilsson et al., 2005*; *Uhlén et al., 2005*). These consortia have generated polyclonal and monoclonal antibodies against proteins and protein domains. Whilst proving successful, in providing a large catalogue of validated antibodies, such efforts have required large multidisciplinary groups across Europe and the US (*Uhlén et al., 2015*). However, the ability to rapidly and cost-effectively generate renewable binding reagents for applications such as studying protein function both in vitro and in vivo and for proteomic projects would represent a major advance. Renewable binding reagents in this context refers to reagents that are recombinantly produced from a known sequence.

The development of alternative binding proteins has provided the opportunity for such advances (*Škrlec et al., 2015*; *Vazquez-Lombardi et al., 2015*). These include reagents such as DARPins (*Binz et al., 2003*), Monobodies (*Koide et al., 1998*), and Affibodies (*Nord et al., 1995*) and a number of others (for a recent review see [*Škrlec et al., 2015*]). Over the past two decades these reagents have proved to be useful tools in many antibody-like applications including detection of proteins for diagnostics (*Theurillat et al., 2010*), for studying protein function (*Kummer et al., 2012*), intracellular targeting of protein function (*Spencer-Smith et al., 2017*; *Wojcik et al., 2010*) and as crystallisation chaperones (*Sennhauser and Grütter, 2008*). In 2010, the University of Leeds and Leeds NHS Teaching Hospital Trust established the BioScreening Technology Group (BSTG), to allow rapid identification of alternative binding proteins against biological targets, particularly those of clinical interest. We now report on some of the outcomes of the more than 350 successful screens performed by the BSTG to date, and suggest that access to this and similar facilities (eg High Throughput Binder Selection facility at the University of Zurich) should deliver the tools needed to complement antibodies in the dissection of biological functions of individual proteins and protein isoforms. Our work is underpinned by the development of a new, engineered protein scaffold for peptide display (*Figure 1*). The Adhiron scaffold is a synthetic protein originally based on a cystatin consensus sequence and displays remarkable thermal stability (Tm = 101°C) (*Tiede et al., 2014*). It is related in structure to a previously reported scaffold engineered from human stefin A (*Stadler et al., 2011*). Binding proteins derived from these two scaffolds are now referred to collectively as Affimer proteins, and we use this term subsequently.

We have previously demonstrated use of Affimers in a number of assays including immune-like (affinity) assays, in biosensors and have tested their ability to be expressed in mammalian cells to manipulate cell signalling (*Tiede et al., 2014*; *Kyle et al., 2015*; *Rawlings et al., 2015*; *Stadler et al., 2014*; *Sharma et al., 2016*). Here we have screened our established Affimer phage library (*Tiede et al., 2014*) against a broad range of targets, including homologous protein family members, to isolate highly specific and renewable binding reagents that can be used both in vitro and in vivo. For broad applicability, and to remove the bottleneck in target protein production, we also tested our ability to generate reagents against small quantities of target protein from commercial sources. We demonstrate the generation of Affimers against various target molecules, including a small organic molecule, and we report their use in a number of widely used biochemical and cell biology assays.

## Results

### Dissecting intracellular signalling pathways

A challenge in cell biology is to develop highly specific tools to detect and modulate the function of one member of a family of structurally and functionally similar proteins. Biological reagents that specifically target a single protein, or a subset of a family of proteins would introduce greater selectivity to in vivo studies.

To demonstrate this functionality, we isolated Affimer binders to various Src-Homology 2 (SH2) domains. SH2 domains are short (~100 amino acid) protein domains that bind specifically to phosphotyrosine-containing motifs in partner proteins, but not to the de-phosphorylated isoforms. They have also recently been found to bind to signalling lipids (*Park et al., 2016*) and are involved in mediating multiple aspects of cellular signal transduction and communication. The human genome encodes some 120 SH2 domains found in 111 proteins (*Liu et al., 2011*). The ability to specifically

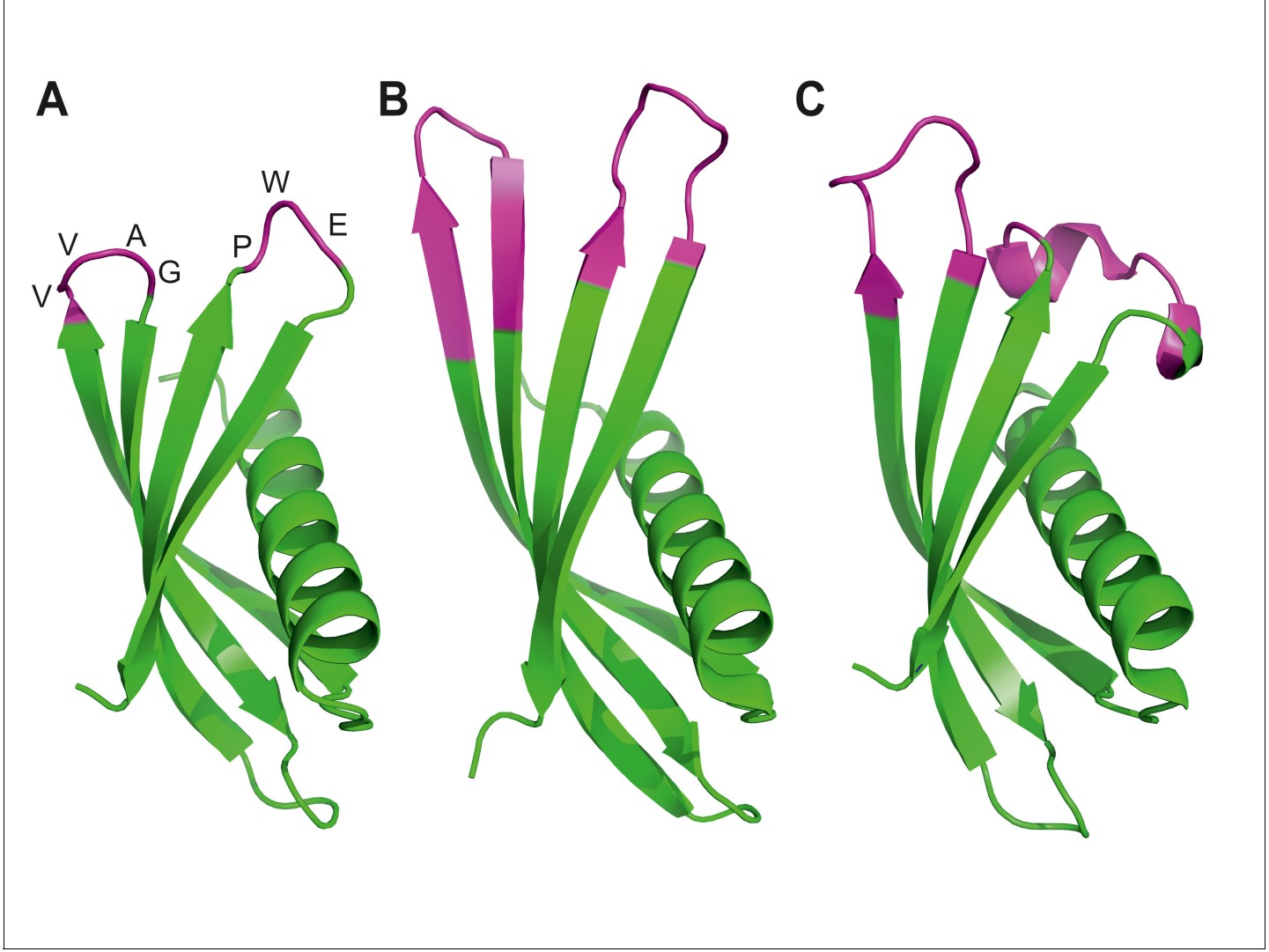

**Figure 1.** Ribbon diagrams of three crystal structures for Affimer (Adhiron) reagents. (**A**) X-ray crystal structure of Affimer scaffold (PDB ID no. 4N6T) at 1.75 A resolution. The amino acids from the loops connecting the four anti-parallel beta sheets are highlighted in pink. (**B**) Crystal structure of an Affimer against p300 (PDB ID no. 5A0O) (**C**) Crystal structure of an Affimer isolated against human SUMO proteins (PDB ID no. 5ELJ). The variable regions in B and C are shown in pink.

detect and inactivate each SH2 domain is a rate limiting step in our understanding of these pathways; the use of siRNA, for example, may be used to remove an entire protein, such as the protein kinases Syk or Zap70, from a cell but will not allow determination of which of the two SH2 domains, carried by each of these kinases, mediate which signalling event(s). The ability to dissect these signalling events with highly specific binding reagents has already identified new biological function using monobodies (*Wojcik et al., 2010*; *Grebien et al., 2011*; *Sha et al., 2013*). We have addressed whether alternative binders can target a specific SH2 domain by selecting Affimers against a range of SH2 domains.

We chose five SH2 domains, some of which had previously been targeted using antibodies (*Renewable Protein Binder Working Group et al., 2011*; *Pershad et al., 2010*). In these previous reports, highly specific binding reagents were identified against the recombinant protein but only a limited number worked efficiently in the tested applications (*Renewable Protein Binder Working Group et al., 2011*). We have previously demonstrated the ability to isolate reagents against the Grb2 SH2 domain (*Tiede et al., 2014*). In the present study we adopted a different target capture

strategy by producing each SH2 domain with an N-terminal biotin acceptor peptide to facilitate simple direct capture from cell lysate and presentation for phage display screening. Each target was checked for efficient biotinylation by Western blot (*Figure 2—figure supplement 1*) and it is noteworthy that this biotinylation was achieved in a BL21(DE3)-derived strain without the need for additional biotin ligase expression. From each screen we randomly selected phagemid clones and by phage ELISA confirmed that Affimers had been selected against each of the nine SH2 domains. The proportion of clones that bound to each target, but not to the control, was between 50% and 100% with an average of 87.6%. Next, we assessed Affimer target specificity by phage ELISA (*Figure 2A*). Grb proteins are growth factor receptor-bound proteins which contain SH2 domains. Initially, the

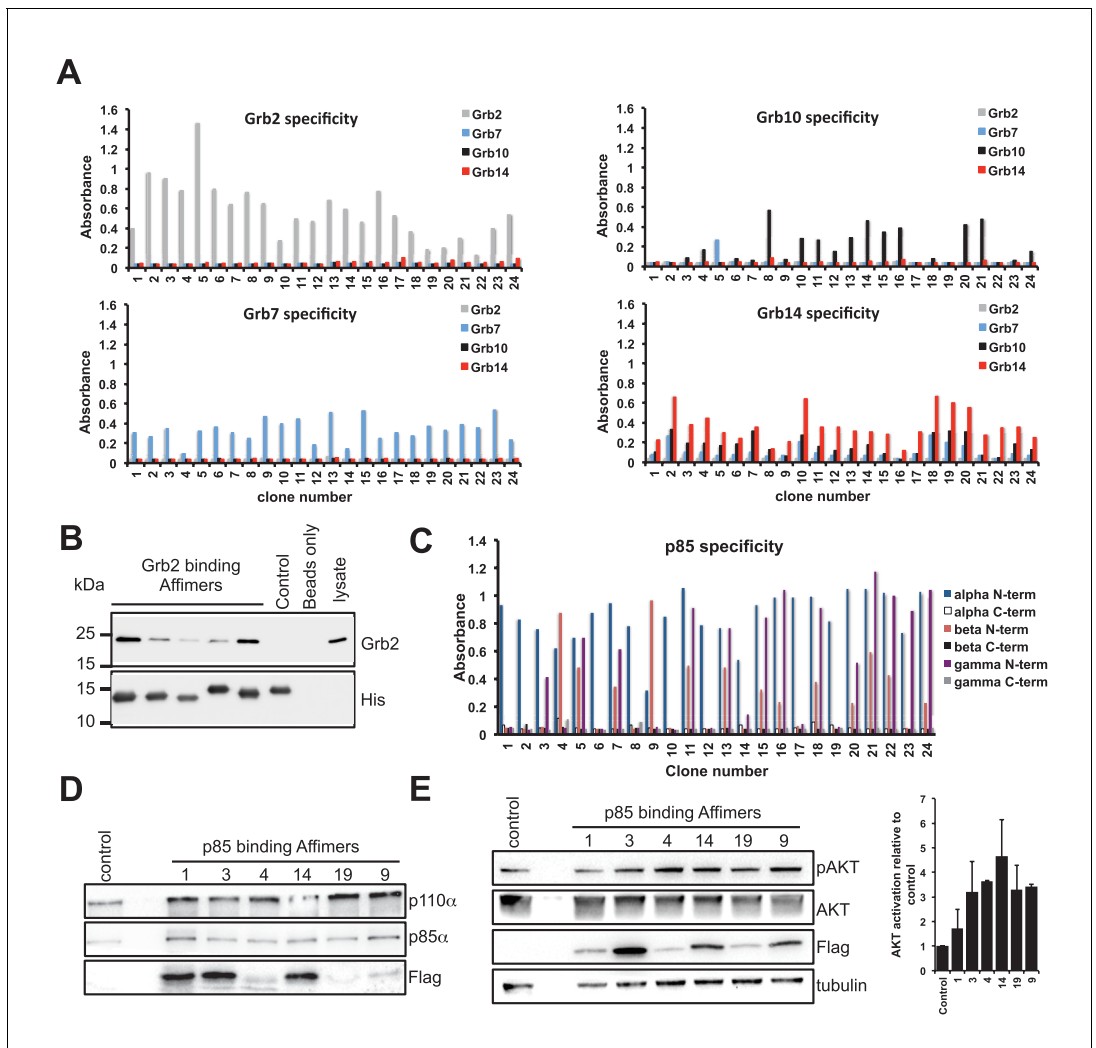

**Figure 2.** Isolation and characterisation of SH2 domain binding Affimers. (**A**) Phage ELISA from 24 monoclonal Affimer reagents isolated against the respective Grb family member SH2 domains. Specificity was tested through extent of binding to the other SH2 family members. (**B**) Western blot showing Affimer-mediated affinity-precipitation of endogenously expressed Grb2 protein from U2OS cell lysates using five Grb2 Affimers bound to colbalt magnetic beads (n = 2). A yeast SUMO binding Affimer was used as a negative control. (**C**) Phage ELISA from 24 monoclonal Affimer reagents isolated against p85 alpha N-terminal domain family member SH2 domain. Specificity was tested through extent of binding to the other p85 SH2 family members. (**D**) Western blot of immunoprecipitation using a p110 antibody on cell lysates from cells expressing p85 SH2 domain binding Affimers (n = 3). (**E**) Western blot and quantification by densitometry of AKT phosphorylation in the presence of expressed p85 SH2 domain binding Affimers (n = 2).

The following figure supplement is available for figure 2:

**Figure supplement 1.** Western blot results of Avi-Tag SH2 domain proteins using an streptavidin-HRP conjugate to detect the presence of biotin.

Grb2, 7, 10 and 14 SH2 domain binding Affimers were tested for cross-reactivity against the other Grb family members, and showed specific binding to the Grb SH2 domain, with the exception of Grb14 Affimers which showed weak cross-reactivity with Grb7 and Grb10 proteins but not Grb2. The level of pairwise sequence homology between Grb7, 10 and 14 is between 65–72% (*Daly, 1998*). It is notable that Affimers were isolated that bind specifically to Grb7 and Grb10 without the need for negative panning to remove cross-reactive binders. We predict that screens that include pre-panning against similar domains would results in isolation of specific Affimers that can bind Grb14 only.

We then examined the ability of Affimer reagents to bind to endogenous proteins. Five of the Grb2-binding Affimers were purified and bound to cobalt-based magnetic beads and their ability to pull down endogenous Grb2 from cell lysates of the human U2OS cell line was assessed (*Figure 2B*, n = 2). All five reagents successfully pulled-down Grb2 whereas a yeast SUMO-binding control Affimer (*Tiede et al., 2014*) was unable to pull down Grb2.

To further assess the ability to isolate isoform specific Affimers we investigated phosphoinositide 3-kinase (PI3K) a heterodimeric protein comprising a p110 catalytic subunit and a p85/p55 regulatory subunit. We examined the specificity of Affimers, raised against the N-terminal SH2 domain of the p85$\alpha$ variant, for cross-reactivity with the p85$\beta$ and p55$\gamma$ variant N-terminal SH2 domains and against the C-terminal SH2 domains of all three isoforms (*Figure 2C*). Despite a high degree of sequence identity (pairwise between 83–90%) a number of p85$\alpha$-specific Affimers were isolated (e.g., clones 1 and 2; *Figure 2C*). Affimers that recognised the $\alpha$ and $\gamma$ but not $\beta$ domain were also isolated (clones 3 and 23). None of these Affimers bound to any of the C-terminal p85/p55 SH2 domains. These results further demonstrate the ability to isolate Affimers that show high binding specificity against related targets, even within a single protein.

The p85 SH2 domain-specific Affimers were expressed in NIH 3T3 cells and their ability to bind to endogenous p85 protein was assessed by co-immunoprecipitation assays, in which p85$\alpha$ was pulled-down. The p85$\alpha$ antibody also pulled down both p110$\alpha$ and the FLAG-tagged Affimers (*Figure 2D*). The different levels of Affimer recovered may be due to differences in Affimer expression levels, as shown in *Figure 2E*, and any differences in binding affinity. It is interesting to note that Affimer 1 appears to bind to endogenous p85 with high affinity (*Figure 2D*) but has little effect on signalling (*Figure 2E*) suggesting it binds outside the key SH2 interaction region. In addition the Affimers did not disrupt the p85/p110 overall complex in which p85 interacts with p110 via three domains, with the SH2 domain regulating activity via binding p110 (*Vivanco and Sawyers, 2002*). These results demonstrate that the Affimer is specifically binding the SH2 domain interaction without affecting the two other p85/p110 binding domains.

We therefore assessed the ability of the Affimers to block the function of the p85 N-terminal SH2 domain by examining whether the Affimers led to an increase in phosphorylated protein kinase B (AKT), a downstream effector of p110. Five of the six Affimer proteins mediated an increase in AKT phosphorylation (*Figure 2E*) demonstrating that they inhibit the interaction between the N-terminal p85 SH2 domain and p110, but importantly do not block p85-p110 complex formation. This supports a report that siRNA inhibition of p85$\alpha$ alone had little effect on cells, but that p85 and p110 both had to be eliminated to produce a phenotype and an effect on AKT phosphorylation (*Kim et al., 2005*). Thus our data highlights a benefit of Affimers, and potentially other alternative reagents, for studying protein-protein interactions within the cellular context.

## Affimers can be used to inhibit extracellular receptor function

Vascular Endothelial Growth Factors (VEGFs) are a family of secreted proteins that regulate many aspects of vascular and lymphatic biology including vasculogenesis (de novo formation of the vascular system), angiogenesis (formation of new capillaries e.g. in response to hypoxia), lymphangiogenesis (de novo formation of the lymphatic system) and arteriogenesis (formation of new arteries e.g. following ischemia). The biological effects of the VEGF family are mediated through binding to a membrane-bound vascular endothelial growth factor receptor (VEGFR) tyrosine kinase subfamily comprising VEGFR1, 2 and 3. While VEGFR1 is implicated as a negative regulator of angiogenesis, VEGFR2 is a major regulator of vasculogenesis, angiogenesis and arteriogenesis. VEGFR3 activation is implicated in specifying lymphangiogenesis but cross-talk between the different VEGFRs can modulate these different processes (*Aspelund et al., 2016*). Dissecting the roles of the different VEGFRs is an important goal, particularly given the success of therapeutic agents targeting VEGF-A in diseases ranging from metastatic cancer to macular degeneration. In this context, VEGFR2 is a key

molecule that regulates many aspects of vascular physiology and blood vessel formation especially angiogenesis and is associated with tumour neovascularisation (*Kofler and Simons, 2015*).

To evaluate whether Affimer proteins that perturb VEGFR2 function could be selected we screened against VEGFR2 and then tested Affimers for their ability to bind recombinant VEGFR2 protein in vitro (*Figure 3A*). In this case DNA sequence analysis revealed that the positive clones represented only two distinct sequences. The affinities for VEGFR2 of representative Affimer proteins, A9 and B8, were determined by SPR to be 41 ± 17 nM and 240 ± 124 nM, respectively (*Figure 3—figure supplement 1*). The Affimer proteins were then labelled at the C-terminal cysteine with a single biotin moiety and used to probe various tissue types for specific staining to compare with the pattern produced by a commercially available polyclonal antibody (*Figure 3B*). The efficiency of Affimer labelling with biotin was determined to be 80–90% by mass spectrometry (data not shown). To directly compare antibody and Affimer patterns a biotinylated secondary antibody was used to detect binding of the primary anti-VEGFR2 antibody. Subsequently, both antibody and Affimer binding were detected by streptavidin-coupled horseradish peroxidase activity. The Affimer reagents showed exactly the same staining pattern as the antibodies, with VEGFR2 staining being predominantly localised in the epithelial cells and with more intense staining at the cell membrane (*Figure 3B*; see arrows). In this case, the staining developed more quickly for the Affimer binders than for the antibody indicating a higher sensitivity of staining.

Although immunohistochemistry is a qualitative rather than quantitative technique, this is an interesting observation given the apparently modest binding affinities, the monomeric nature, and the mono-biotinylated state of the Affimer binders compared to the bivalent nature of, multiply biotinylated polyclonal antibody molecules. It demonstrates the value of Affimers as affinity histo-chemistry reagents. The differential sensitivity of staining may be due to the difference in size between antibody and Affimer with the latter better able to penetrate the fixed tissue more efficiently. The Affimer may also have a more exposed binding site compared to the antibody. One or more of these properties may allow a greater number of binding events to the target resulting in higher sensitivity of Affimer staining.

Alternative scaffolds have been reported to inhibit VEGFR2 including Nanobody (*Behdani et al., 2012*), Adnectin (*Tolcher et al., 2011*), Affibody (*Fleetwood et al., 2014*) and DARPin (*Hyde et al., 2012*) proteins, so we questioned whether the Affimer proteins could also inhibit VEGFR2 signalling in human vascular endothelial cells (HUVECs). Previous siRNA studies (*Murga et al., 2005*) have shown that VEGFR2 signalling is required for the formation of vascular tubules by transfected HUVECs, although this siRNA-mediated effect requires 24–48 hr following transfection. By contrast, the inhibitory effect of Affimer B8 could be measured within just 30 min of treatment and also led to a decrease in VEGF-dependent tubule length and branch point formation in a tubulogenesis assay (*Figure 3C*). Consistent with the effects on tubulogenesis, Affimer B8 also inhibited VEGF-dependent phosphorylation of VEGFR2 and downstream signalling, with decreased activation of cell signalling mediators PLCg1, AKT, ERK, p38 and eNOS (n = 3; *Figure 3D*). By contrast control Affimers had no effect on signalling. Overall these observations demonstrate that Affimers represent useful research reagents that are capable of blocking the biological function of specific receptors on biologically-relevant timescales.

## Affimer binders for modulating ion channel function

Ion channels are involved in a number of physiological processes, and are important drug targets (*Overington et al., 2006*). However, there remains a lack of reagents able to modulate ion channels with the selectivity and specificity required to prevent off-target effects (*Skerratt and West, 2015*). Antibodies have proven to be useful as ion channel imaging reagents and have recently shown promise as therapeutics (*Lee et al., 2014*; *Sun and Li, 2013*). Complementing the repertoire of antibodies available, smaller biologics are increasingly being used to study ion channels, for example, by providing crystallization chaperones (*Stockbridge et al., 2015*; *Zhou et al., 2001*). Furthermore, the high selectivity often associated with such biologics alongside their ability to access functional crevices may provide further opportunities to modulate ion channel function. Indeed, the targeting of both ligand and voltage-gated ion channels by Nanobodies and scFv's, respectively, has already demonstrated this potential (*Danquah et al., 2016*; *Harley et al., 2016*).

Here, we set out to isolate Affimers capable of binding to and modulating the activation of the Transient Receptor Potential Vanilloid 1 (TRPV1) ion channel by screening against a peptide derived

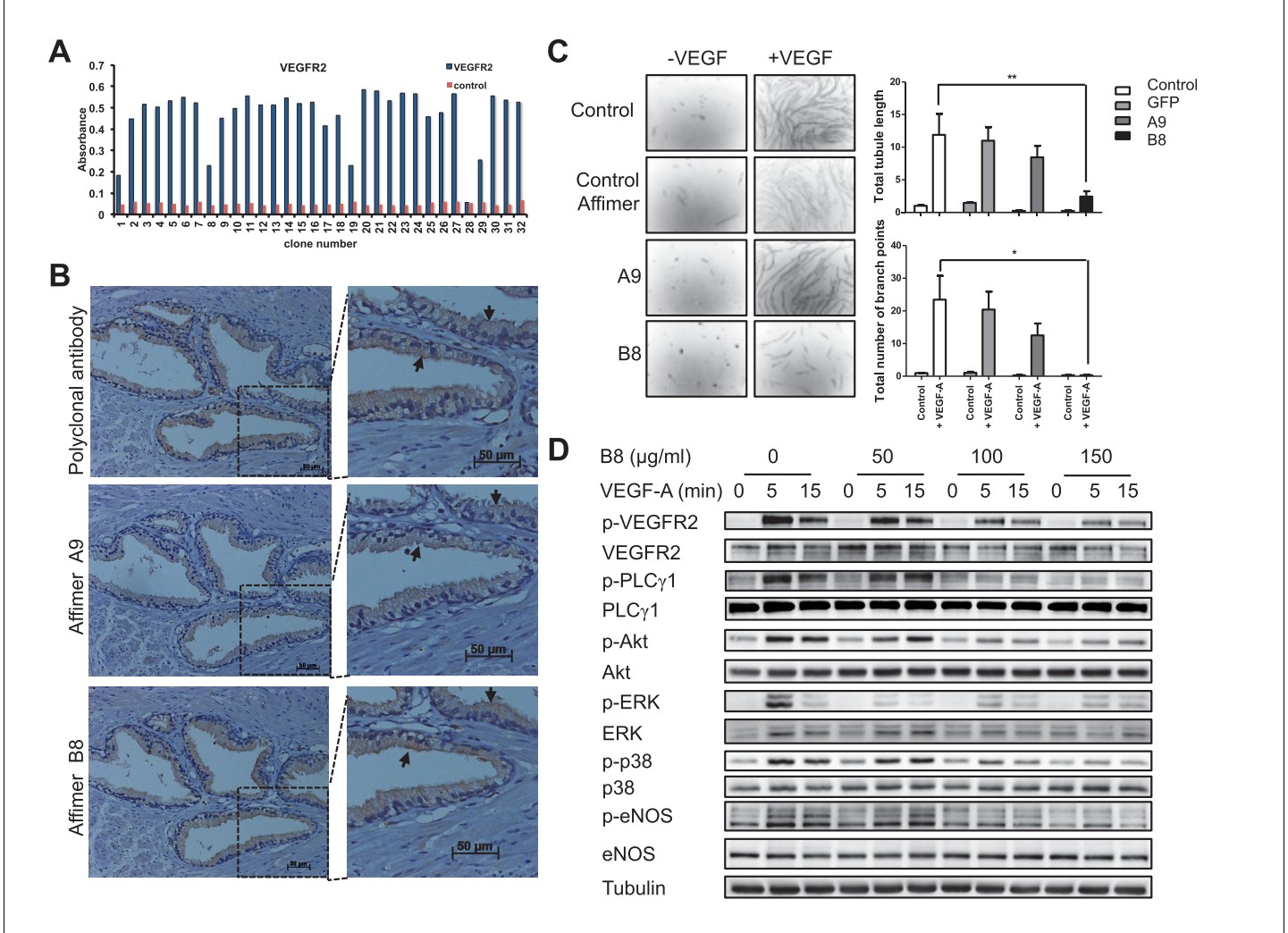

**Figure 3.** Characterisation of VEGFR2 binding Affimers. (**A**) Phage ELISA for 32 monoclonal Affimer reagents isolated against VEGFR2. The negative control contained just streptavidin. (**B**) Immuno- and affinity-histochemistry of a polyclonal anti-VEGFR2 antibody and of representative Affimers B8 and A9. Staining is shown as a light brown color, haemotoxylin counter staining (blue). Arrows show similar staining patterns. (**C**) Tubulogenesis assay in the presence and absence of vascular endothelial growth factor A and the two Affimers with quantification of tubule length and branch point number shown to the right. The control is in the absence of any Affimer and the control Affimer is a binder against yeast SUMO (n = 3). Statistical analysis was performed using a two-way ANOVA followed by the Bonferroni multiple comparison test using GraphPad Prism software (La Jolla, USA). *p* values less than 0.05 (*), 0.01 (**) are indicated on the graphs. Error bars in graphs denote ± standard error of mean. (**D**) Western blot results showing changes in downstream signalling in HUVECs treated for 0, 5 and 15 min in the presence of vascular endothelial growth factor A and increasing concentrations of the VEGFR2 binding Affimer B8 (n = 3).

The following figure supplement is available for figure 3:

**Figure supplement 1.** SPR plots for the anti-VEGFR2, TNC and TNT binding Affimers.

from the outer pore domain. Thirteen unique Affimer clones were identified from 24 positive clones identified by phage ELISA of 96 randomly selected colonies from the phage library screen (*Figure 4A*). None of the 13 binders showed cross-reactivity to a distinct peptide derived from the pore region of a voltage-gated sodium channel, Nav1.7. Affinity-fluorescence studies were performed to examine the ability of the Affimer proteins as detection reagents. Only Affimer 2 stained U2-OS cells expressing full-length TRPV1 (*Figure 4B*) showing co-localisation with an anti-TRPV1 antibody (*Figure 4C*). Affimer 2 showed no staining of TRPV1-negative U2-OS control cells. None of the other 12 binders worked in this assay.

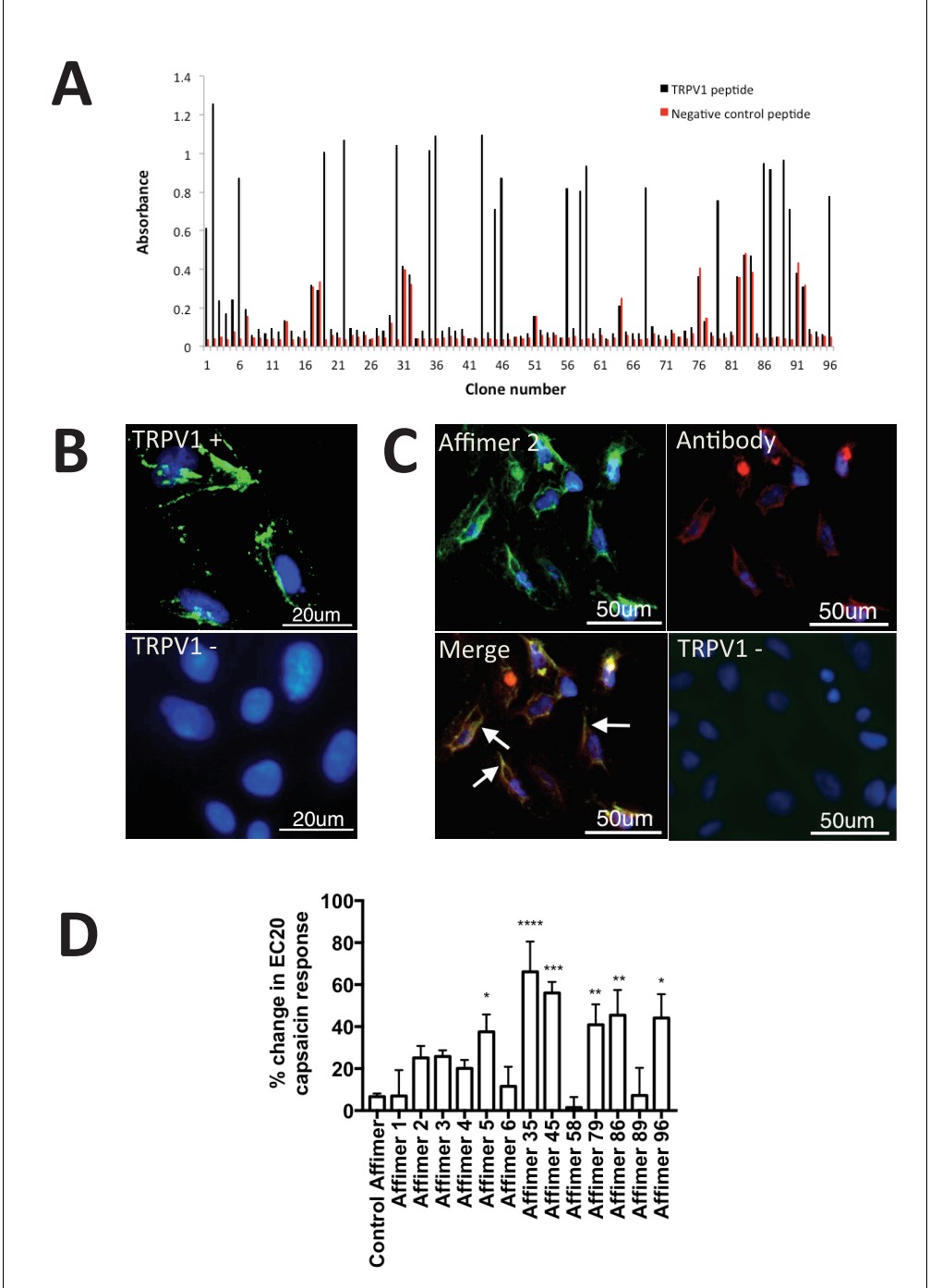

**Figure 4.** Characterisation of TRPV1 binding Affimers. (**A**) Phage ELISA for 96 monoclonal Affimer reagents isolated against TRPV1 peptide. The negative control contained a different hydrophobic peptide sequence. (**B**) Affinity-cytochemistry on U2-OS cells transiently transfected with TRPV1 (TRPV1+) or control (TRPV1-) using Affimer 2. Binding was detected using an anti-HIS antibody fluorescently labeled with FITC. Binding of the Affimer is shown as a green and DAPI (a DNA stain) shown as blue (n = 3), (**C**) Co-localisation of Affimer staining with an anti-TRPV1 antibody. Antibody staining is shown in red. (**D**) A Flexstation was used to measure uptake of Fluo-4 AM, a calcium binding fluorescent small molecule, to measure calcium levels in capsaicin stimulated cells in the presence of Affimer control and TPRV1-binding Affimers (n = 3).

Next we investigated TRPV1 modulation by measuring the levels of intracellular calcium in response to treatment with the Affimer proteins. While no direct modulation was observed six Affimers showed significant enhancement of TRPV1 activation upon treatment with the agonist capsaicin (*Figure 4D*) compared with cells treated with capsaicin alone. Previous research has explored the use of small molecule compounds as positive modulators of TRPV1 to desensitize and reduce pain (*Kaszas et al., 2012*). The compound (MRS1477) was hypothesised to interact with the pore-forming region of TRPV1, leading to a three-fold increase in capsaicin activation when applied at low micromolar concentrations – an effect similar to that reported here for some of the Affimers. Overall, this study demonstrates that Affimer proteins can be raised against a peptide surrogate to recognise and alter ion channel function by positive allosteric modulation, a suggested mechanism for the treatment of TRPV1-induced chronic pain (*Lebovitz et al., 2012*) and may represent a novel approach and therapeutic strategy for chronic pain relief.

## Affimer binders for in vivo imaging

Tenascin C (TNC) is an extracellular matrix protein that is abundant during early development, is expressed at low levels in adult tissues and is frequently up-regulated in cancer tissues and associated with metastasis (*Minn et al., 2005*; *Oskarsson et al., 2011*) and poor patient outcomes (*Lowy and Oskarsson, 2015*). As such, it offers potential as a tumour marker for imaging and/or therapeutic targeting in vivo (*Hicke et al., 2006*). Affimer binders to TNC were isolated from the phage display library (*Figure 5A*). One Affimer protein with high affinity for TNC ($K_D$ = 5.7 ± 2.8 nM by SPR – sup *Figure 2*) was used in subsequent assays. To evaluate its specificity for TNC we compared the staining pattern of the Affimer to that of an anti-TNC antibody in human colorectal cancer and glioblastoma xenograft tissue sections. Staining patterns with C-terminally biotinylated TNC Affimer were similar to those obtained with the TNC antibody (*Figure 5B*).

In the clinic, tumour imaging using labelled antibodies can be limited by a high background of circulating labelled antibody until this is cleared from the body. This leads to extended hospital stays or the necessity for multiple patient visits for a single test. By contrast, the smaller size of alternative binding proteins means that the molecules that do not bind to the target, will be more quickly cleared from the circulatory system, more conveniently allowing visualisation shortly after imaging agent administration. To demonstrate this in tumour bearing mice we visualised distribution of TNC Affimer compared to a control GFP-binding Affimer, both C-terminally labelled with Rhodamine Red (*Figure 5C*). To maximise signal detection, we imaged excised tumours and organs post-sacrifice and quantitated the signal as fold-change above background. As expected, both Affimer probes were detected in kidney indicating renal clearance. However, compared to the TNC Affimer this clearance was faster for the GFP Affimer as it showed a significant decrease (p=0.04) in fluorescence from 24 to 48 hr post-injection (fold change 9.23 ± 3.10 at 24 hr to 2.98 ± 0.77 at 48 hr; *Figure 5D*). The TNC Affimer signal at 24 hr post-injection was significantly higher (p=0.02) in tumours (6.26 ± 1.62) compared with the control GFP Affimer group (2.32 ± 0.61) suggesting that the TNC Affimer accumulated in the TNC expressing tumour. The TNC probe was also detected in liver tissues either due to hepatobiliary clearance or due to the fact that TNC shows low level expression in normal liver sinusoids (*Van Eyken et al., 1990*). In addition, the ratio of anti-TNC Affimer binders in tumour compared to the spleen, for example, was >6 at 24 hr (*Figure 5D*); in contrast, anti-TNC antibodies took 2 days to reach a tumour/spleen ratio of 5, although this did improve to 20–30 at day 10 (*De Santis et al., 2006*). Thus the more rapid clearance rate of alternative binding proteins, such as Affimers, compared to antibodies has the potential to allow more rapid imaging of tumours. Further work to enhance signal detection in vivo with Affimers is underway (*Fisher et al., 2015*).

## Affinity-fluorescence in fixed cells

Marek's Disease, caused by Marek's Disease Virus (MDV-1), is a globally and economically significant neoplastic disease of chickens that is currently controlled by vaccination with the related Herpes Virus of Turkeys (HVT). In field samples, tests for Marek's Disease would need to be able to distinguish between proteins from HVT and their homologues in MDV. As a proof of principle we screened the phage library against HTV-derived protein UL49, with counter screens against host proteins as well as the related proteins MDV (RB1B) and DEV UL49. Phage ELISA, affinity-fluorescence and in-cell Western confirmed that the selected Affimers were specific for HTV recombinant proteins

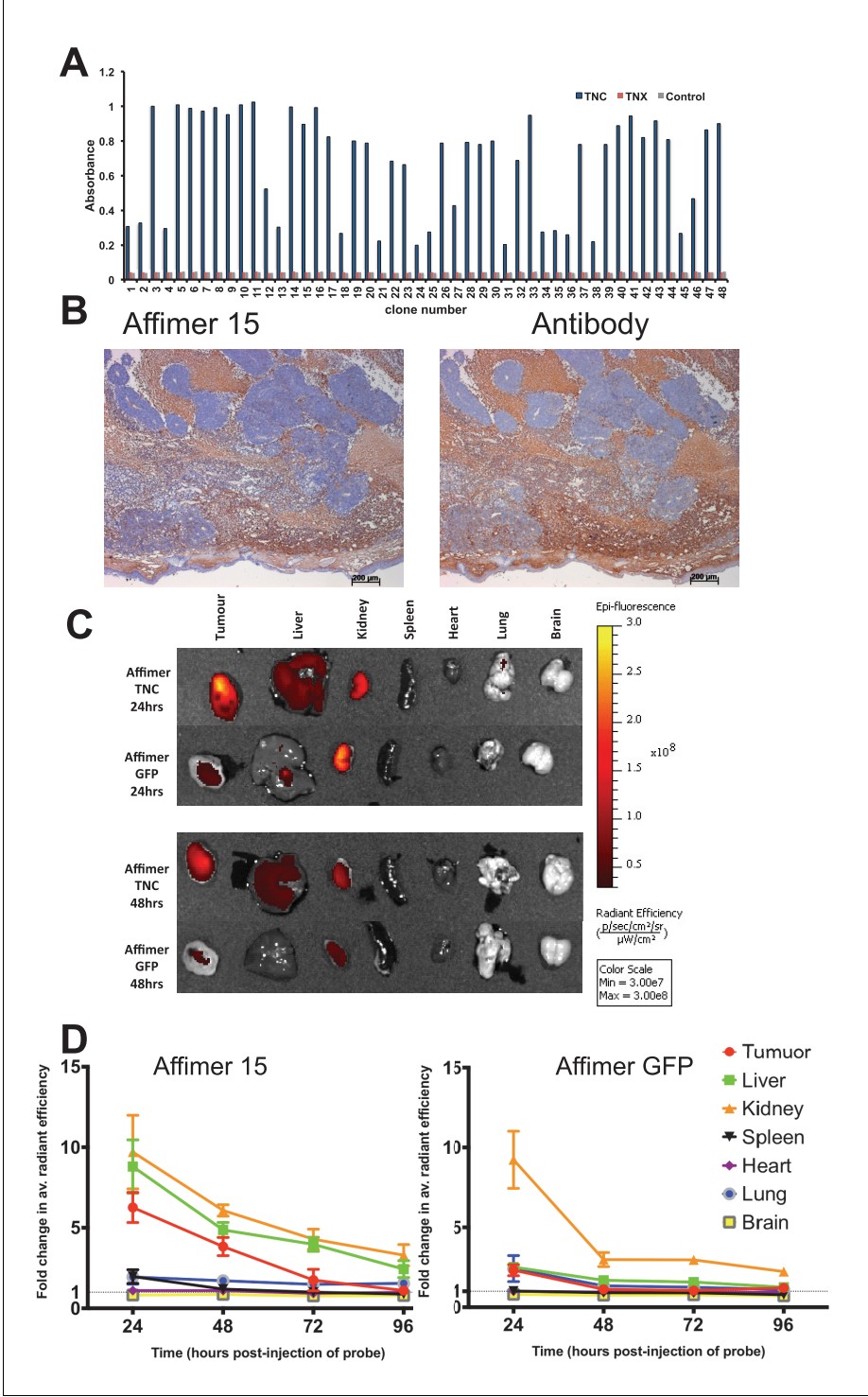

**Figure 5.** Characterisation of tenascin C (TNC) binding Affimer by affinity-histochemistry and ex vivo imaging of xenografts. (**A**) Phage ELISA for 48 monoclonal Affimers against TNC. The two controls are tenascin X (TNX) and streptavidin. (**B**) Immunohistochemistry of serial sections of a mouse xenograft (SW620 cell line), showing staining for TNC. Antibody/Affimer staining is shown as a light brown color with haemotoxylin counter staining (blue). (**C**) and (**D**) Mice were injected via their tail vein with rhodamine labelled TNC binding Affimer or a control GFP binding Affimer. After 24, 48, 72 and 96 hr the xenograft and organs were removed and visualized. (**C**) Organ images at 24 hr. (**D**) Quantification of rhodamine fluorescence (radiant efficiency in $p/s/cm^2/sr/\mu W/cm^2$) ex vivo (n = 3). Mean background fluorescence intensity was normalized to sham injected control tumors and organs.

as well as their ability to specifically stain the target protein in primary Chicken Embryonic Fibroblasts (CEFs) containing bacterial artificial chromosome (BAC) clones of MDV-1 (RB1B), HVT or DEV (strain 2085) (*Figure 6A,B and C*).

The affinity of anti HVT UL49 Affimers was in the low nM range, $K_D$ values of 1.5 nM to 7.5 nM with a mean 3.3 nM, for the eight clones tested (*Figure 6—figure supplement 1*). The high specificity and affinity should be advantageous in the development of DIVA (differentiating infected from vaccinated animals) tests for the discrimination of vaccine and field strain viruses. We tested the performance of anti-HVT UL49 Affimers in affinity-fluorescence (*Figure 6C*). Cultures of primary CEFs infected with HVT GFP BAC clone were subjected to affinity-fluorescence staining using biotinylated anti HVT UL49 Affimers and were visualised with streptavidin-Alexa Fluor 568 conjugate (*Figure 6C*). Compared with low background staining with the streptavidin only control (SA only) there are pronounced cytoplasmic foci detected by Affimers in infected cells. These foci are consistent with data from the related alphaherpesvirus MDV (*Denesvre et al., 2007*; *Rémy et al., 2013*) or the model alphaherpesviruses Herpes Simplex type 1 (HSV-1) (*Stylianou et al., 2009*) and Pseudorabies virus (PrV) (*del Rio et al., 2002*) and likely indicate the cytoplasmic sites of HVT secondary viral envelopment. This distribution is also consistent for the different Affimer clones tested and is seen only within infected cells. Thus Affimers show promise as alternatives to traditional antibodies and are likely to be particularly valuable where availability/performance of existing antibody reagents is poor.

## Affimers as probes for super-resolution microscopy and single particle tracking

Super-resolution microscopy provides the ability to localise proteins within a cell at ca. 20 nanometer resolution. A major limitation of wide-spread exploitation of this approach is the lack of highly specific reagents that can place the fluorophore in close proximity to the endogenous target protein. Antibodies are large multi-domain proteins that are normally labelled with fluorophores at random sites that limits the achievable resolution. By contrast, the smaller Affimer proteins can be labelled in a site-specific manner providing closer spatial placement of the fluorophore to the target protein, thus facilitating use of current super-resolution techniques. This approach has recently been demonstrated using Nanobodies where super-resolution microscopy was used to image GFP-tagged proteins and nuclear pore complex (*Pleiner et al., 2015*; *Ries et al., 2012*).

Human epidermal growth factor receptor 4 (HER4), also known as c-erbB-4, is an oncogenic transmembrane receptor protein kinase (*Lemmon and Schlessinger, 2010*). Although the function of this protein is not yet fully understood, it is known to be associated with increased survival and lower proliferation in breast cancer patients (*Machleidt et al., 2013*). We screened the phage display library against HER4 (*Figure 7A*) and two Affimers were recombinantly produced with a C-terminal cysteine for labelling with the fluorophores Alexa Fluor 647 or CF640R maleimide. The Affimer showing the highest signal by fluorescent imaging was used for further studies. Our results show that the HER4 Affimer can bind both to CHO cells transiently expressing HER4 and to MCF7, a breast cancer cell line expressing lower physiological levels of HER4 (*Figure 7*). When HER4 is over-expressed in CHO cells the Affimer showed increasing binding at concentrations from 5 nM to 100 nM, as determined by membrane signal intensity from confocal microscopy images, while in MCF7 cells that express physiological levels of HER4 binding increases from 10 to 200 nM Affimer (*Figure 7B*).

Dual colour wide field fluorescence images of HER4 receptor fused at the intracellular C-terminal end with eGFP (HER4-CYT-eGFP) in CHO cells (*Figure 7C* - top) and labelled in the extracellular region with Affimer-Alexa 647 (*Figure 7C* middle) shows that the HER4 Affimer can be used to specifically label membrane-associated HER4, through co-localisation of GFP and labelled Affimer fluorescence. The corresponding direct Stochastic Optical Reconstruction Microscopy (dSTORM) image (*Figure 7C* – bottom) has a localisation precision of ca. 25 nm. A Bayesian cluster analysis (*Griffié et al., 2016*) of the dSTORM image shows that the most prevalent cluster size of HER4 oligomers is between 8.1 and 12 nm in radius. This corresponds with the most prevalent number of HER4 molecules in a cluster being between 4 to 8. These data show that HER4 forms oligomers as large as those previously found in EGFR. (*Needham et al., 2016*)

This Affimer is also suitable to image HER4 under Total Internal Reflection Fluorescence (TIRF)-mode to undertake single-particle tracking on live cells (*Figure 7D*). To detect single particles, the binding affinity of the Affimer must be in the low nM range to avoid saturating the sample and to

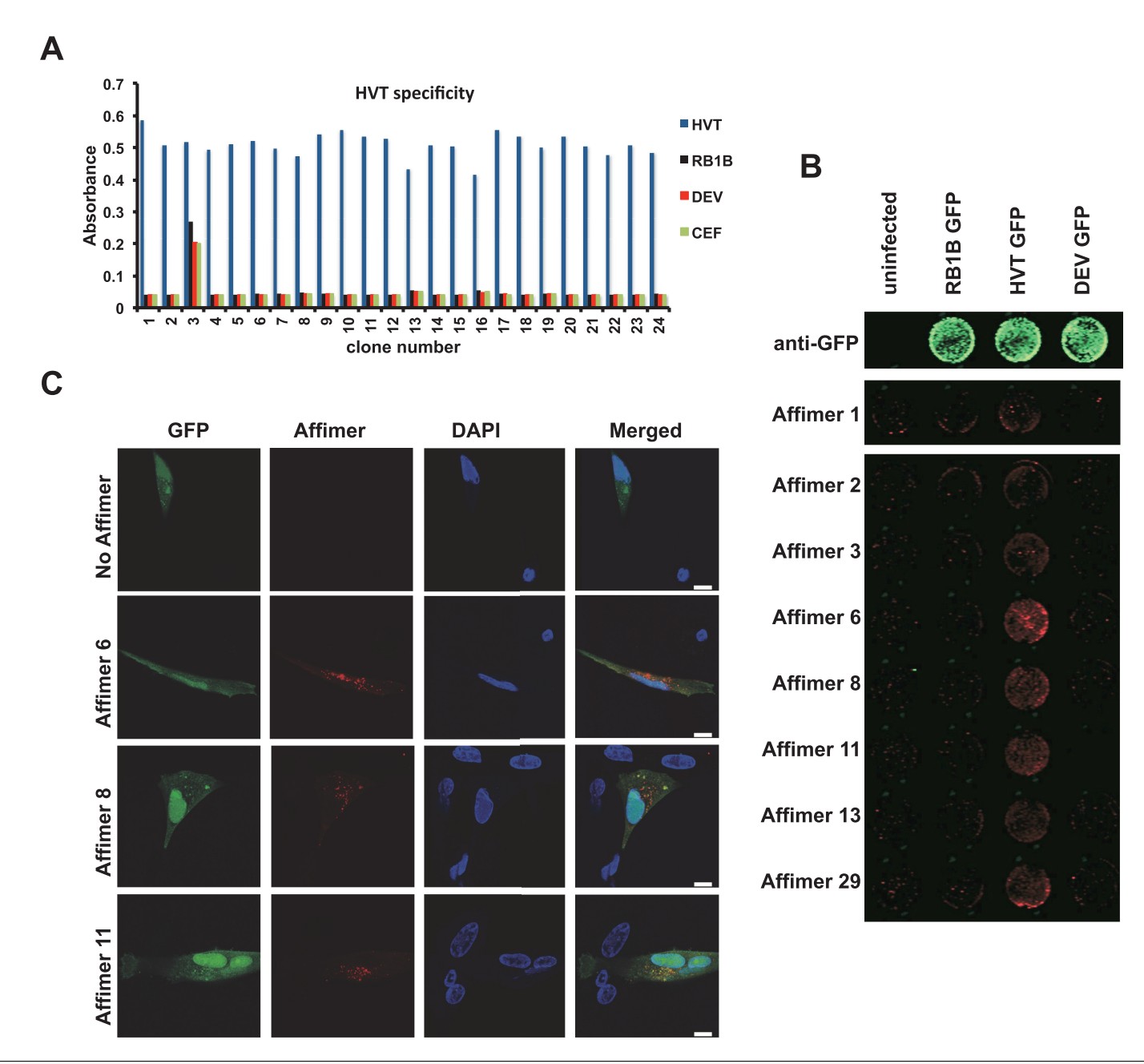

**Figure 6.** Affimer detection of HVT UL49 in infected cells by in-cell Western and affinity-fluorescence. (**A**) Phage ELISA for 24 monoclonal Affimers against HVT screened against HVT, RB1B, DEV and CEF lysates to confirm specificity for HVT. (**B**) Infection of cells was confirmed using a goat anti-GFP antibody and donkey anti-goat 680 (green) antibody to detect GFP which is constitutively expressed by the BAC derived viruses. Infected CEFs were screened with candidate HVT UL49 Affimers at 1.5 μg/ml with subsequent detection (red) by Streptavidin 800 conjugate (Licor). (n = 3) (**D**) HVT infected CEFs were screened with HVT UL49 Affimers and visualised with Streptavidin-568 (red). Nuclei were stained with DAPI (blue). Streptavidin only control (no Affimer) shows no observable labelling of infected CEFs. Bar = 10 μm.

The following figure supplement is available for figure 6:

**Figure supplement 1.** Bilayer Interferometry plots for the HVT binding Affimers.

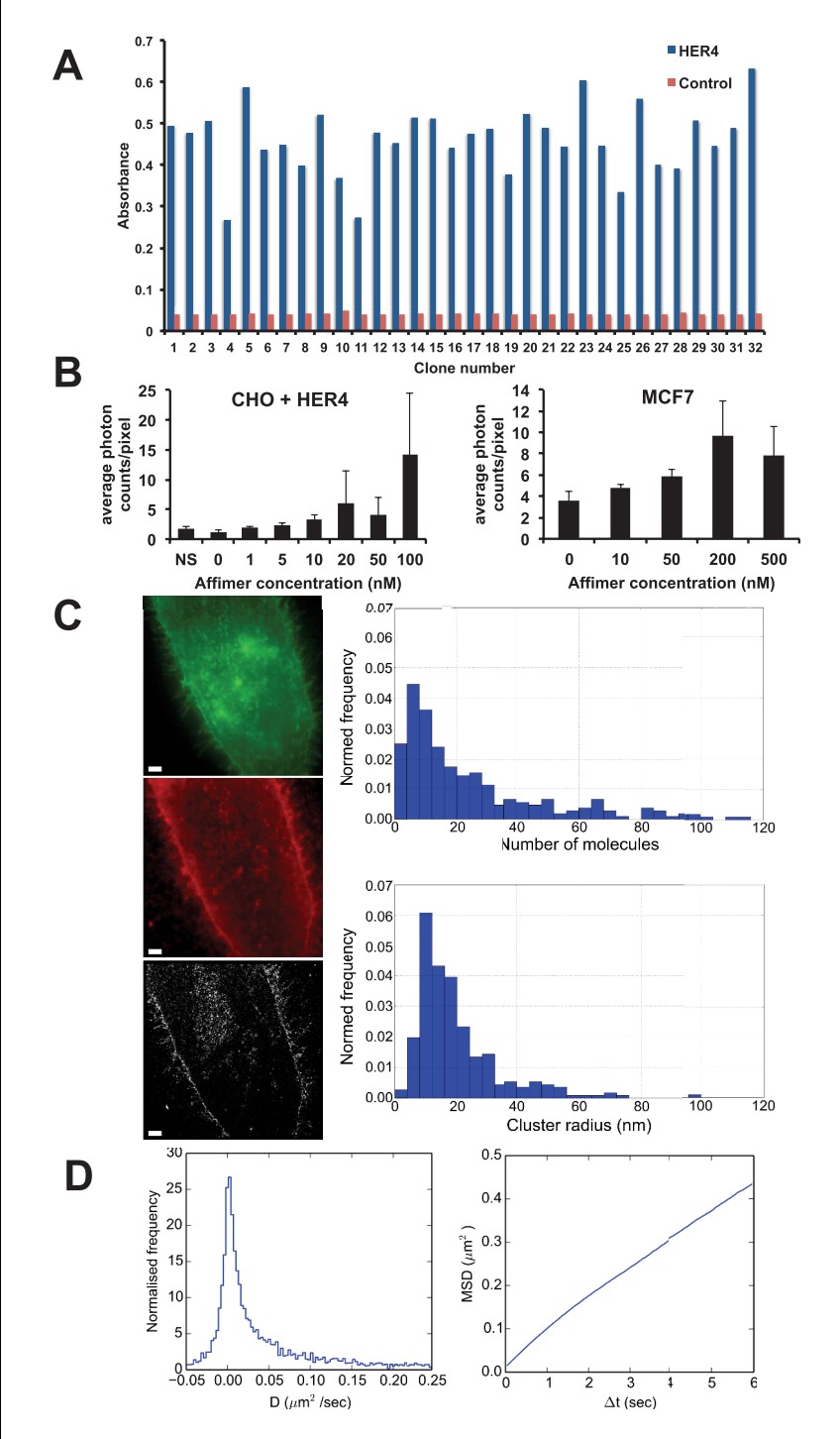

**Figure 7.** Use of HER4 binding Affimers in super-resolution imaging and single molecule tracking. (**A**) Phage ELISA for HER4 binding Affimers. (**B**) Average photon counts/pixel for HER4-binding Affimer labelled with CF640R and bound to CHO cells transfected with HER4 and to MCF7 cells expressing endogenous levels of HER4. (**C**) Wide field image of CHO cells transfected with HER4-CYT-eGFP showing localisation of HER4 via GFP fluorescence (top) and labelled with HER Affimer–Alexa647 (middle). The corresponding dSTORM image of HER4 Affimer conjugated to Alexa647 (bottom) with a 25 nm localisation precision. Scale bar = 2 µm. Right plots to show the number of molecules and cluster size of clusters identified by dSTORM. (**D**) Diffusion coefficients (left panel), and MSD curve (right panel) of HER4 Affimers labelled with CF640R and tracked on MCF7 cells expressing endogenous HER4.

reduce non-specific binding. HER4 particles were tracked with a Bayesian tracking algorithm (*Rolfe et al., 2011*) and the diffusion coefficient and Minimum Square Displacement (MSD) were calculated from the resulting trajectories. The data show that there is an immobile, or scarcely mobile, population of HER4 receptors on MCF7 cells, associated with a tail of highly mobile molecules (*Figure 7D*, left panel). The near straight slope of the MSD plot indicates that, unlike EGFR (*Needham et al., 2016*; *Zanetti-Domingues et al., 2012*), the diffusion of HER4 is not confined on the timescales investigated (*Figure 7D*, right panel).

The Affimers raised against HER4 demonstrate the ability to isolate reagents that can be used in a range of super-resolution microscopy techniques. However, as there is no direct comparison to an antibody this example does not highlight the advantage of alternative proteins over the larger antibody probes. To provide this demonstration Affimers have also been raised against polymerised microtubules (*Figure 8*). The Affimer we selected labels interphase microtubules in a similar way to a widely-used antibody (*Figure 8A*). However, in mitotic cells, the Affimer labels the spindle but not astral microtubules (*Figure 8A*) likely reflecting the fact that the antibody recognises tyrosinated microtubules unlike the Affimer. Interestingly, the Affimer is able to label the central region of the cytokinetic furrow (*Figure 8A*), in which microtubules are very densely packed. Antibodies are usually excluded from this region so analysis of this feature has been problematic (*Hu et al., 2012*). This highlights one advantage of using smaller probes, such as alternative binding proteins, for super-resolution microscopy and for example in this case will allow further elucidation of the role tubulin plays in cytokinetic furrows.

3D dSTORM images of microtubules using the antibody and Affimer look similar (*Figure 8B*) with analysis showing that Affimer labelling has the expected central decrease in fluorescence for binding to the outside of the microtubule (*Figure 8C*). However, averaging profiles for multiple microtubules for both Affimer and antibody shows the increased localisation accuracy with Affimers, compared with antibodies (*Figure 8D*). While localisation density may not be fully optimised in these samples (*Huang et al., 2008*) the average microtubule profiles were substantially narrower with Affimer labelling (47 ± 11 nm) than with primary antibody labelling (73 ± 10 nm) (FWHM, mean ± s.d.) and should allow further elucidation of tubulin structures that have previously not been solved. Overall, Affimers, and presumably other alternative binding proteins, have an advantage over antibodies in labelling for dSTORM.

## Affimers can be selected against small organic compounds

The generation of effective binding reagents to low molecular mass organic compounds is technically challenging. Small molecules do not display innate immunogenicity and thus are typically conjugated to carrier proteins to elicit an effective immune response. Even so it can be a problem raising an immune response to toxic molecules and those that conjugate poorly to carrier proteins. To examine whether we could isolate Affimer reagents against a small organic molecule we used 2,4,6-trinitrotoluene (TNT). Previous studies have shown that presentation of TNT as a hapten for antibody production is known to be vital for the successful isolation of TNT specific antibodies (*Ramin and Weller, 2012*). The TNT analogue 2,4,6-trinitobenzene sulphonic acid (TNBS) (*Figure 9A*) contains nitro-groups ($NO_2$) located in the same positions as TNT on the benzene ring, while the methyl ($CH_3$) group is substituted by a sulfonic acid ($SO_2OH$) group. This functional group reacts with primary amines and was used to prepare both TNBS-ovalbumin and TNBS-IgG conjugates for phage display screening, with counter screens performed against ovalbumin and IgG to enrich for small molecule binding.

To confirm binding specificity selected clones were tested against both TNBS conjugates and unconjugated protein by phage ELISA (*Figure 9B*). The number of clones that showed strong binding to TNBS conjugated with both ovalbumin and IgG was relatively high (22/32) with a further 8/32 showing reasonable binding. Of the 32 clones tested 14 distinct sequences were identified, some containing only variable region 1 which indicates selection from a small sub-population of the original phage display library. Within variable loop 1 it was possible to define a short consensus sequence implying a common binding mode. Four Affimer proteins were purified and tested for binding to TNT and various dinitrotoluenes (DNT; *Figure 9A*) by competition ELISA (*Figure 9C*). All four Affimers showed binding to the original TNBS-conjugate and to TNT, but differed in their specificity for the DNTs. Affimer 4, a VR1 only Affimer, showed a higher level of specificity for TNT than any of the DNTs. By contrast, Affimer 3 binds to TNT and also provides discrimination between 2,4-DNT

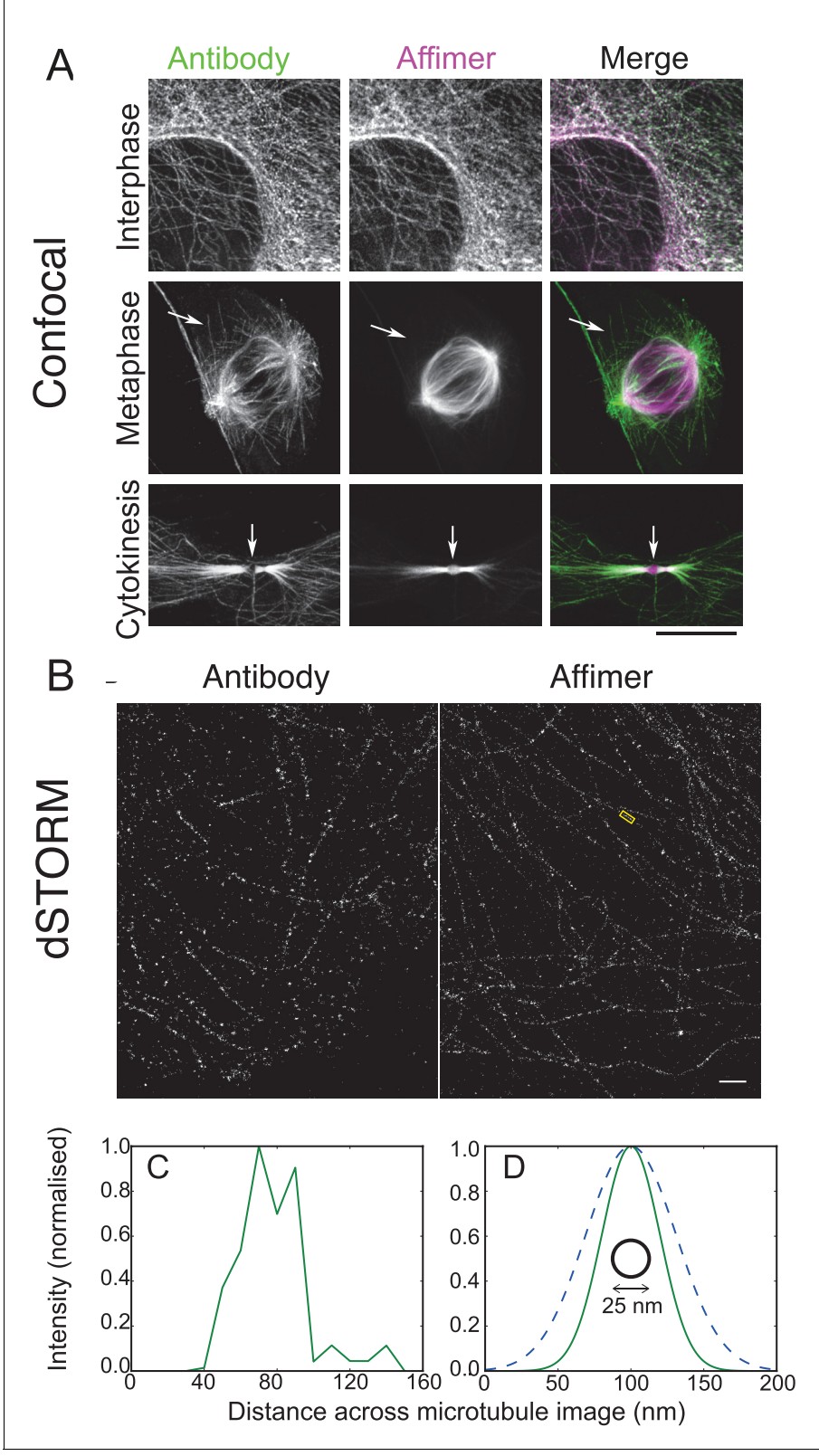

**Figure 8.** Use of tubulin binding Affimer in super-resolution microscopy. (**A**) Confocal images of microtubules in HeLa cells, stained with a rat α-tubulin antibody (YL1/2) which recognises tyrosinated tubulin, and an Affimer for polymerised tubulin, conjugated to Alexa Fluor 647. Images of an interphase and metaphase cell, together with an image of the cytokinetic furrow are shown. Arrows in the metaphase cell point to astral microtubules that are predominantly labelled with the antibody. Arrows in the cyokinetic furrow indicate the central region (Fleming body). Scale bar is 10 μm. (**B**) 3D

*Figure 8 continued on next page*

*Figure 8 continued*

dSTORM images of microtubules in a HeLa cell, labelled with Alexa Fluor 647 conjugated to a primary antibody to rat α-tubulin (left) and an Affimer for polymerised tubulin (right). These images are from separate cells. Localisations were aggregated into 10 nm bins and projected onto a single plane, with Gaussian smoothing. Scale bar 1 μm. (C) Intensity profile across the microtubule image labelled in (B) (yellow box), averaged along 510 nm of its length. The central decrease in intensity reflects the hollow structure of the microtubule. (D) Comparison of the average microtubule image intensity profile with antibody staining (dashed, mean of 6 microtubule sections), Affimer staining (solid, mean of 8 microtubule sections) and actual microtubule size (black circle). The FWHM of each average profile (as in (C)) was found for a Gaussian fit and a Gaussian distribution is plotted here using the mean FWHM for each staining method.

and the other two DNT's. This high selectivity of recognition of the nitro group on position 4, demonstrates that Affimer reagents can show remarkable specificity for such small molecular differences. It is highly likely that altering the panning strategy to include competition steps with analogues would allow selection of specificity and sensitivity of Affimers for small organic molecule targets.

The ability of Affimer proteins to bind to small molecules raises the possibility that they may be used in cells to quench the effects of molecules such as Shield or doxorubicin that are currently used to regulate protein behaviour, allowing investigators to assess the effects of switching protein interactions off with the same speed with which they are currently turned on.

## Discussion

The ability to rapidly isolate highly specific alternative binding protein affinity reagents that perform consistently in a wide range of scientific applications is the 'holy grail' for producing renewable binding reagents. For a recently developed artificial binding protein scaffold, known as Affimer, we demonstrate such applicability across a range of molecular and cellular studies. We have isolated Affimer proteins against more than 350 targets, but here we have exemplified their use as molecular and cellular tools against 12 different target molecules. Typically each screening regime, consisting normally of three panning rounds of phage display and phage ELISA, was normally completed in 12 days. Thus Affimer selection, as with other alternative protein or antibody fragment selection approaches that use phage display or other in vitro selection, is much faster than antibody and nanobody production techniques that involve animal inoculation. They also allow the efficient identification of binding reagents against conformational epitopes since target proteins are screening in their folded state.

Each selected Affimer coding region was sub-cloned into an *E. coli* expression vector and recombinant protein was purified over a further seven days. Without automation the phage display platform allows an individual to screen up to 24 targets simultaneously. The benefit of such manual screening is that greater control can be exercised to moderate individual screening regimes within a set of samples. A further advantage of Affimer proteins is the ability to express recombinant protein at high yield in *E. coli*. Of the 36 Affimers reported on here an average yield of 83.3 mg/L (1.5–188 mg/L) culture was achieved with a purity of greater than 95% following a single immobilised metal affinity step. We have not attempted to optimise the level of protein production but typically only grow 50 mL cultures for protein purification that provides suitable quantities of protein for most applications.

The key to successful isolation of Affimers for cellular studies is the use of high-quality antigens normally presented via biotin/streptavidin on plates and beads. Recombinant sources of protein are normally of high quality as time and effort is taken to purify the protein. For some recombinant proteins, particularly those being expressed in mammalian cells, this can be more challenging. We expressed the small SH2 domains with an N-terminal biotin acceptor peptide to facilitate site-specific in vivo biotinylation allowing target immobilisation onto streptavidin plates, directly from cell lysate (*Figure 2—figure supplement 1*). This approach should have wide applicability for recombinant protein and domain target presentation for screening protocols. For the commercially sourced protein antigens (tenascin C, VEGFR2, tubulin and HER4) it became apparent that the success of the screen is dependent upon the commercial source. For example, tenascin C was sourced from several commercial suppliers but only one allowed the selection of suitable Affimers (data not shown). Fortunately the potential availability of thousands of high quality proteins from structural genomics

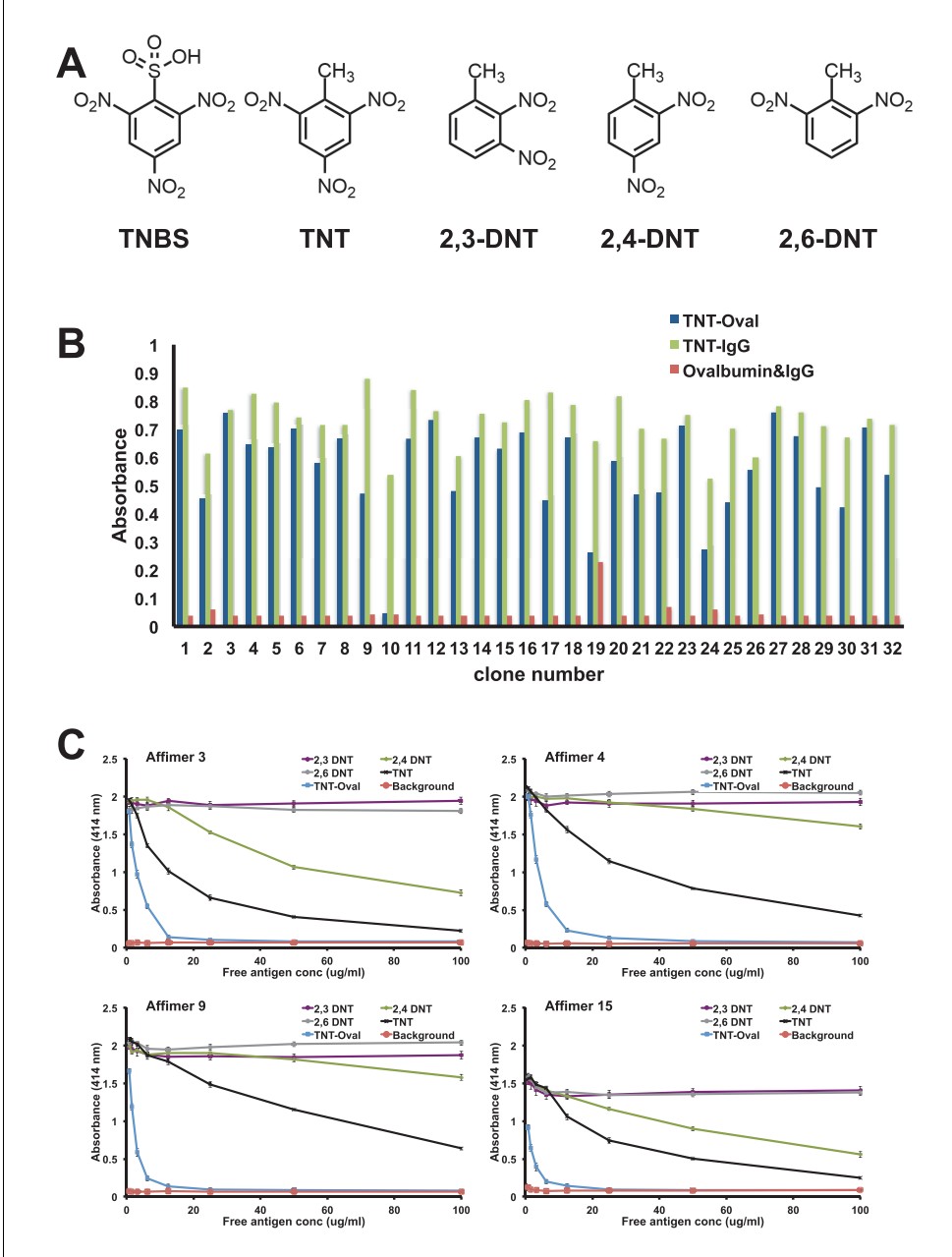

**Figure 9.** Affimer selection and specificity against TNT and DNT's. (**A**) Chemical structures of TNBS, TNT, 2,3-DNT, 2,4-DNT and 2,6-DNT. (**B**) Phage ELISA results from 32 monoclonal Affimer reagents isolated against TNBS bound to ovalbumin. Binding specificity was also tested against TNBS bound to IgG and to unconjugated ovalbumin and IgG. (**C**) Competition ELISA of four TNT-Affimers to check specificity against a range of molecules across a concentration profile. Error bars = standard deviation from technical repeats of a representative ELISA.

consortia together with the ability to express in vivo biotinylated protein domains for capture, without purification, should reduce the risk of screen failure associated with target quality and availability.

Two papers have described the generation of antibody and antibody fragments against SH2 domains (*Renewable Protein Binder Working Group et al., 2011*; *Pershad et al., 2010*). The initial paper used phage display to select binding reagents that showed exquisite specificity against recombinant SH2 domains in vitro (*Pershad et al., 2010*). However, none of the reagents were used

in assays to demonstrate binding to endogenous proteins from cells. Colwill et al assessed the ability to isolate binding reagents against the same family of targets using both phage display of antibodies fragment libraries and monoclonal antibodies (*Renewable Protein Binder Working Group et al., 2011*). They also successfully isolated binding reagents, although only a low proportion bound endogenous protein in the assays tested. By contrast, with the same class of SH2 targets, a high proportion of the monoclonal Affimer reagents reported here were successful in pull down assays and could block protein function when expressed in cells. The differences between these outcomes may be a result of library quality rather than an inherent feature of the scaffold. However, we consider it more likely due to differences in presentation of variable loop structures between antibodies and the Affimer scaffold.

The ability to express Affimers intracellularly in mammalian cells, as shown by the inhibition of the p85 SH2 domain (*Figure 2*), represents an exciting opportunity, with the lack of disulfide bonds in the scaffold suited to the reducing environment of the cell. This feature is similar to other artificial binding proteins that can also be expressed in the cytoplasm of mammalian cells (*Kummer et al., 2012*; *Spencer-Smith et al., 2017*; *Wojcik et al., 2010*). This raises the intriguing possibility of generating the necessary reagents, based on Affimers and other artificial binding proteins, to target specific protein domains of the human 'interactome'. These would provide extremely powerful tools for understanding the function of proteins and for identifying novel drug targets in disease. The lack of disulphide bonds in the Affimer and many other artificial proteins, such as DARPins, Monobodies and Affibodies, also allows the directed introduction of cysteine residue(s) for site specific chemical modification, including addition of a single biotin or fluorophore.

A major advantage of phage display screening is the ability to isolate highly specific reagents by performing counter-screens against very similar target molecules. Interestingly for the SH2 targets, no counter-screening was performed and yet by ELISA analysis, specific Affimers were recovered for the Grb2, 7 and 10 and p85 SH2 domains. Further studies will determine whether this level of specificity is observed at the cellular level. These results, nonetheless, provide promise for the isolation of highly specific cellular binding reagents. This high level of specificity was also demonstrated with Affimers that bind the small organic compound TNT commonly used as a model organic compound. The Affimers revealed remarkable specificity considering the small size of the molecule (mol. mass <300 Da) and the limited number of panning rounds used. It would be interesting to determine molecular structures of Affimer bound to TNT and 2,4-DNT to understand the recognition mechanism and to explain the discrimination between different DNT molecules. The ability to rapidly select Affimers that specifically detect small molecule targets represents a useful additional approach to the generation of reagents for diagnostic and monitoring applications of chemical agents, for example, in health, security or environmental settings. There are many examples of antibodies that bind to small molecules, although the in vivo nature of raising such reagents can present challenges for some compounds, such as toxins and pharmaceuticals, together with the time frame for inoculation and isolation and the need to use animals. Phage display has also been used to isolate antibody fragments (*Dörsam et al., 1997*; *Vaughan et al., 1996*) and lipocalins that recognise small molecules (*Beste et al., 1999*; *Schlehuber et al., 2000*).

The selection of reagents against small molecules also raises the prospect of recognising a range of post-translational modifications, potentially within the context of a specific protein. For example, DARPins that discriminate between phosphorylated and non-phosphorylated proteins have been described (*Kummer et al., 2012*). These recognise conformational changes due to the phosphorylation event rather than the phosphorylated amino acid. Only time will tell whether alternative binding reagents are capable of directly reporting on post-translational modification of proteins in a similar manner to antibodies.

The binding affinities of Affimers selected in this work were typically in the low nanomolar range, although some of the VEGFR2 binders had weaker affinities. Even so, these weaker binders still worked effectively and displayed specificity in affinity-histochemical assays and inhibited receptor function in biological assays. Since our monoclonal reagents have been identified by randomly selecting clones from three panning rounds it is anticipated that inhibitors with greater affinity can be developed, either through more detailed analysis of the pool of phage using next generation sequencing, or by affinity maturation. Whilst our screening strategy identifies monoclonal reagents, these can also be combined to generate polyclonal reagents that may improve sensitivity for certain in vitro applications.

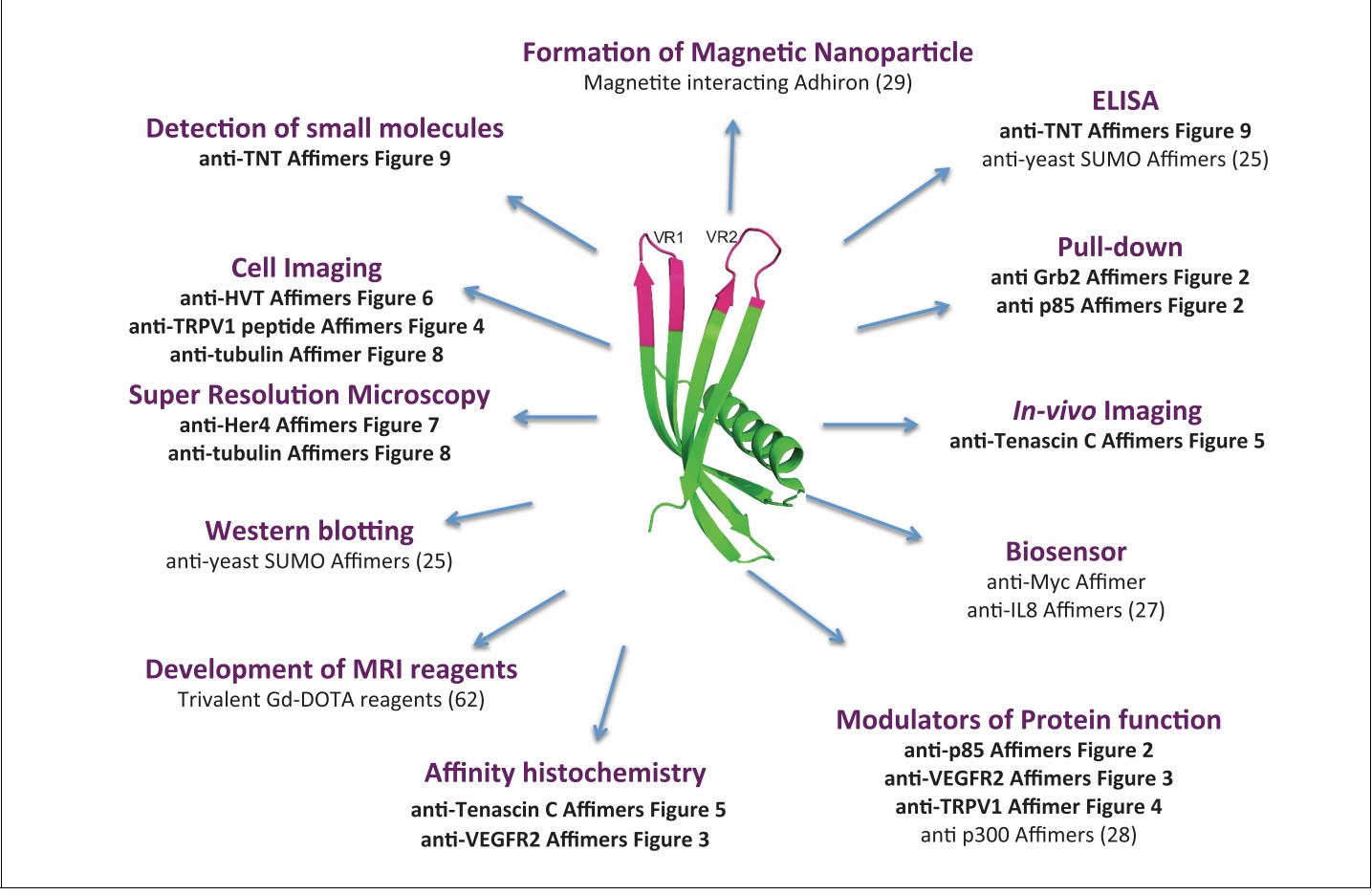

**Figure 10.** Overview scheme of a range of applications that have been tested with Affimers.

In conclusion, we have demonstrated the ability to rapidly isolate Affimer reagents, that are effective tools in a range of molecular and cell biology applications (*Figure 10*). This highlights the potential for creating a pipeline to isolate consistent renewable binding reagents against a wide variety of target molecules. Affimers are small, thermostable and simple to engineer and provide a system that compliments rather than replaces antibodies and other alternative protein scaffolds. A major aim of our laboratory is to further explore the capabilities of Affimer reagents and to test their potential protein modulating properties for use in dissecting specific cell signalling pathways as well as in studying protein-protein interactions on a proteomic scale. Affimer technology is commercially available through Avacta Life Sciences or for academic collaborations through the University of Leeds. Alternatively, the library can be synthesised as described by *Tiede et al. (2014)* and screened in individual laboratories, making this technology immediately accessible to the scientific community.

## Materials and methods

### Expression and purification of targets

Human SH2 domain coding sequences were in kanamycin-resistant pET28 SacB AP vectors (Open Biosystems), with an N-terminal histidine tag. A biotin acceptor peptide (BAP) sequence was cloned into the vector to give an N-terminal BAP-Histag-SH2 domain sequence and the modified vector DNA introduced into Rosetta 2 (DE3) cells. Single colonies were grown in 10 ml Terrific Broth (TB) supplemented with 100 µg/ml kanamycin and 34 µg/ml chloramphenicol, overnight at 37°C and 2 ml was used to inoculate 400 ml TB/100 µg/ml kanamycin and cultures grown until $OD_{600}$ ~2. After cooling to 18°C, for 1 hr IPTG was added to 0.5 mM. Cells were collected by centrifugation (800 *g*;

20 min, 4°C) and resuspended in 10 ml Lysis buffer 1 (50 mM NaH$_2$PO$_4$; 300 mM NaCl; 30 mM imidazole; 10% glycerol; Benzonase Nuclease (Novagen); 1% Halt Protease Inhibitor Cocktail, EDTA-free; 1% Triton-X100; 1% lysozyme) and left rocking overnight at 18°C before target proteins were purified using Amintra Ni-NTA resin (Expedeon). Proteins were eluted using elution buffer (50 mM NaH$_2$PO$_4$; 500 mM NaCl; 300 mM imidazole; 10% glycerol). Expression and in vivo biotinylation of targets was confirmed by western blotting.

The *UL49* gene of Herpesvirus of Turkeys (HVT) was amplified by PCR using Q5 DNA polymerase (NEB, UK) and cloned using the Gibson Assembly kit (NEB) into a modified pMT-V5/6His (Invitrogen) in which the V5/6His cassette was replaced with 6His-AviTag and a SmaI site that was used to generate an N-terminal fusion to the UL49 gene product. pMT HVT UL49 was co-transfected with pCoHygro (Invitrogen) into Drosophila S2 cells using calcium phosphate precipitation and stably transformed cells were selected with hygromycin according to the manufacturer's instructions (Invitrogen) before expression testing by western blot analysis 24 hr after induction with 500 µM copper sulphate. For purification, protein was extracted from stably transformed cells 36 hr after induction with copper sulphate at 20°C using a modified lysis buffer (25 mM Tris (pH8), 1.5% Triton-X100, 50 mM arginine (pH8), 10 mM imidazole, 7.5% glycerol, 300 mM KCl). Protein was eluted from Ni-NTA (QIAGEN) in elution buffer (25 mM Tris (pH8), 50 mM arginine (pH8), 200 mM imidazole, 7.5% glycerol, 300 mM KCl). Purified protein was subjected to in vitro biotinylation using purified BirA according to the manufacturer's instructions (Avidity LLC).

## Preparation of 2,4,6-trinitrobenzene protein conjugate

A 2,4,6-trinitrobenzene protein conjugate was prepared by mixing ovalbumin (fraction VI, Sigma) or Rabbit Gamma Globulin (RGG) (Sigma) at a concentration of 1 mg ml$^{-1}$ with 0.05% (w/v) 2,4,6-trinitrobenzene sulfonic acid (TNBSA) (Thermo Scientific) in 0.1 M sodium bicarbonate buffer (pH 8.5). The mixture was incubated at 37°C for 2 hr and the resultant complex was then buffer exchanged into PBS and concentrated using a Vivaspin six column (MWCO: 10 kDa; Sartorius) to eliminate unconjugated TNBSA and excess buffer. The concentrated product was quantified using a Pierce Micro Bicinchoninic Acid (BCA) Assay (Thermo Scientific), in accordance to manufactures guidelines using bovine serum albumin as the comparative standard protein.

## Phage display

Target biotinylation and selection of Affimers by phage display was performed as described previously (*Tiede et al., 2014*) with some modifications in the second and third panning round. Biotinylated targets were bound to streptavidin-coated wells (Pierce) for 1 hr, then 10$^{12}$ cfu pre-panned phage (phage preincubated in streptavidin-coasted wells) were added for 2.5 hr with shaking. Panning wells were washed 10 times and phage eluted with 50 mM glycine–HCl (pH 2.2) for 10 min, neutralised with 1 M Tris–HCL (pH 9.1), further eluted with triethylamine 100 mM for 6 min, and neutralised with 1 M Tris–HCl (pH 7). Eluted phage were used to infect ER2738 cells for 1 hr at 37°C and 90 rpm then plated onto LB agar plates with 100 µg/ml carbenicillin and grown overnight. Colonies were scraped into 5 ml of 2TY medium, inoculated in 25 ml of 2TY medium with carbenicillin (100 µg/ml) and infected with ca. 1 × 10$^9$ M13K07 helper phage. After 1 hr at 90 rpm, kanamycin was added to 25 µg/ml for overnight at 25°C and 170 rpm. Phage were precipitated with 4% polyethylene glycol 8000, 0.3 M NaCl and resuspended in 1 ml of 10 mM Tris, pH 8.0, 1 mM EDTA (TE buffer). A 2 µl aliquot of phage suspension was used for the second round of selection using streptavidin magnetic beads (Invitrogen). Target labelled beads were washed and incubated with pre-panned phage for 1 hr then washed five times using a KingFisher robotic platform (ThermoFisher), incubated overnight at RT in 20% glycerol in PBS-T with at least one additional wash step and eluted and amplified as above. The final pan used neutravidin high binding capacity plates (Pierce), as previously described for panning round one with the addition of a final overnight incubation at RT in 20% glycerol in PBS-T, and phage eluted using 100 µl of 100 mM dithiothreitol. Phage eluates were recovered from wells containing target protein and control wells to determine the level of amplification in target wells. For counter selections appropriate cell lysates or homologous proteins were added to the phage at a concentration of at least 10 µg/ml for 1 hr at room temperature before transferring the phage to the panning beads or wells. Counter screens were performed for UL49 target against purified Marek's Disease Virus type 1 (MDV-1, strain RB1B) and Duck enteritis virus (DEV

strain 684), as well as cell lysate derived from Chicken Embryonic Fibroblasts (CEF). The TNBS-Ovalbumin was counter selected against ovalbumin.

In the second and third pan, after washing the wells/beads post incubation with the phage library, the wells/beads were incubated over night at RT in 20% glycerol in PBS-T with at least one additional wash step prior to elution. Phage were eluted in 100 µl 50 mM glycine–HCl (pH 2.2) for 10 min, neutralised with 15 µl 1 M Tris–HCl (pH 9.1), further eluted with 100 µl triethylamine 100 mM for 6 min, and neutralized with 50 µl 1 M Tris–HCl (pH 7). Phage ELISA screening was performed, as previously described, on randomly selected clones from the final pan round as a method for selecting positive clones for further evaluation (*Tiede et al., 2014*).

## Affimer protein production

Selected Affimer coding regions were amplified by PCR amplification using one of two reverse primers that would generate proteins with or without a C-terminal cysteine. Following NheI/NotI digestion the coding regions were ligated into a pET11a-derived vector and subsequently expressed in BL21 (DE3) cells as previously described (*Tiede et al., 2014*). Briefly, a single colony was used to inoculate a 5 ml overnight culture in 2TY/100 µg/mL carbenicillin. Then 50 ml LB-carb media was inoculated with 1 ml of overnight culture and grown for about 2 hr at 37°C and 230 rpm to an $OD_{600}$ between 0.6–0.8, before addition of IPTG to 0.1 mM and further grown for 6–8 hr or overnight at 25°C at 150 rpm. Cells were harvested, lysed in 1 ml Lysis buffer. The lysate was then incubated with 300 µl of washed NiNTA slurry for 1 hr, washed (50 mM NaH2PO4, 500 mM NaCl, 20 mM Imidazole, pH 7.4) and eluted in 50 mM (NaH2PO4, 500 mM NaCl, 300 mM Imidazole, 20% glycerol, pH 7.4).

## Biotinylation of Affimers

Affimers with C-terminal cysteine were biotinylated directly after elution. For each Affimer 150 µl tris (2-carboxyethyl)phosphine (TCEP) immobilised resin (ThermoFisher Scientific) was washed and incubated with 150 µl of 40 µM Affimer solution on a rocker for 1 hr. The solution was centrifuged for 1 min at 1500 *g* and 120 µl of the supernatant was transferred into a fresh tube containing 6 µl of 2 mM biotin-maleimide (Sigma) and incubated for 2 hr at room temperature. Excess biotin linker was removed by using a Zeba spin desalting column (Thermo Scientific) according to the manufacturer's protocol or by dialysis.

## Immunoprecipitation and western blot analysis

Western blotting was performed on expressed BAP-tagged SH2 domains to check in vivo biotinylation of the targets. Protein samples were re-suspended in 4 X sample buffer (8% (w/v) SDS, 0.2 M Tris-HCl (pH 7) 20% glycerol, 1% bromophenol blue) and heated to 95°C for 10 min. Samples were loaded onto a 15% SDS-polyacrylamide resolving gel with a 5% stacking gel. Electrophoresed samples were then transferred onto a PVDF membrane with 0.2 µm pore size using a Trans-Blot Turbo Transfer System (Bio-Rad, Hercules, USA). Membranes were blocked in 3% BSA in Tris buffered saline (TBS) containing 0.1% Tween-20 (TBS-T) overnight at 4°C followed by incubation with High Sensitivity Streptavidin-HRP (ThermoFisher Scientific) and visualised using Luminata Forte Western HRP Substrate (Merck Millipore).

Cell lysates for pull down of Grb2 were isolated from U-2 OS cells (purchased from ATCC, STR profiled and mycoplasma negative - RRID:CVCL_0042). Cells were washed with ice-cold 1 X PBS and lysed in lysis buffer (50 mM Tris, 150 mM NaCl, 1% (v/v) Nonidet P-40, 1 ml per 75 $cm^2$ flask), on ice. The pull down experiment was performed using the KingFisher automated platform. In brief, Affimers were expressed in 50 ml BL21 Star (DE3) IPTG induced overnight culture, the BL21 Star (DE3) cells were pelleted, lysed and clarified by centrifugation. Cell lysate was incubated with cobalt-based magnetic beads (ThermoFisher) on Kingfisher platform for 10 min prior to a single wash, and incubated with U-2 OS cell lysate (approximately 500 µg) for 90 min. The beads were washed a further three times on the KingFisher prior to being added to 50 µl of elution buffer. A 15 µl aliquot of eluted proteins was mixed with loading buffer prior to being heated to 95°C for 5 min and loaded on to an SDS-PAGE gel. Western blot analysis was performed using a rabbit monoclonal anti-Grb2 antibody, (Abcam; ab32037) as primary antibody and anti-rabbit-HRP antibody, goat polyclonal (Signal Technology; 7074) as secondary (n = 2, biological replicate; the number of times the experiment was independently repeated).

For p85 pull down experiments NIH-3T3 cells transfected with p85-nSH2 Affimers were lysed in CellLytic M Cell Lysis Reagent (Sigma) with protease inhibitor cocktail (Sigma) and lysates cleared by centrifugation. For AKT activation western blot analysis, cells were serum-starved for 30 min prior to protein harvesting. Immunoprecipitation of p85$\alpha$ involved incubating 400 µg of total protein with 3 µl of anti-p85$\alpha$ antibody (Abcam) at 4°C with rotation overnight followed by protein G Sepharose beads (Sigma) for 4 hr. The beads were washed four times with PBS and resuspended in 2 X SDS sample buffer with $\beta$-mercaptoethanol (Sigma), heated to 95°C for 3 min and proteins were resolved in 7% SDS-polyacrylamide gels. For western blotting, 20 µg of total protein was denatured using 5 X Laemmli sample buffer with $\beta$-mercaptoethanol and resolved in 12% SDS-polyacrylamide gels. Proteins were transferred to polyvinylidene difluoride membranes (Biorad), blocked in 5% bovine serum albumin in PBS 0.1% Tween, and incubated with anti-pAKT (Ser473), anti-panAKT, anti-p110$\alpha$ (Cell Signaling Technology), anti-flag M2 (Sigma), anti-tubulin alpha (AbD Serotec), and anti-p85$\alpha$ antibodies. Bound antibody was detected using anti-mouse/rabbit horseradish peroxidase-conjugated antibody and chemiluminescence (Luminata Forte Western HRP substrate, Millipore).

For detection of VEGFR2 signalling HUVECs (primary human umbilical vascular endothelial cells) were lysed in 2% (w/v) SDS containing 1 mM PMSF, in PBS). For immunoblot analysis, lysates were incubated at 95°C for 5 min and sonicated for 3 s. 25 µg of cell lysate was re-suspended in an equal volume of 2 X SDS sample buffer (1 M Tris-HCl pH 6.8, 20% (v/v) glycerol, 4% (w/v) SDS and 0.1% (w/v) bromophenol blue, 4% (v/v) mercaptoethanol) and incubated at 95°C for 5 min. Lysates were loaded onto a 10% (v/v) SDS-polyacrylamide resolving gel with a 5% (w/v) SDS-polyacrylamide stacking gel and separated at 130 V for 90 min in SDS-running buffer (192 mM glycine, 25 mM Tris, 0.1% (w/v) SDS). Proteins subjected to SDS-PAGE were transferred onto nitrocellulose membrane (0.2 µm pore size) (Schleicher and Schuell) in transfer buffer (106 mM glycine, 25 mM Tris, 0.1% (w/v) SDS, 20% (v/v) methanol) at 300 mA for 3 hr at 4°C. Membranes were incubated in 5% (w/v) skimmed milk (in TBS-T) for 30–60 min on a rocker. Membranes were rinsed in TBS-T, incubated with primary antibodies (goat anti-VEGFR2 (R&D Systems), rabbit anti-phospho-VEGFR2 (Y1175), rabbit antibodies to native and phosphorylated PLC$\gamma$1 (Y783), c-Akt (S473), p38 MAPK (T180/Y182) and eNOS (S117), rabbit anti-ERK1/2, mouse anti-phospho-ERK1/2 (T202, Y204) (Cell Signalling Technologies), mouse anti-$\alpha$-tubulin (Santa Cruz Biotechnology)) overnight at 4°C and washed 3 times for 10 min in TBS-T prior to incubation with HRP-conjugated secondary antibody (PerBio Sciences, Cramlington, UK) for 1 hr at room temperature, followed by a second round of TBS-T washes and detection using a chemiluminescent solution, EZ-ECL (Geneflow, Nottingham, UK).

## Immuno- and Affinity-fluorescence for TRPV1

U2-OS cells were grown in DMEM medium supplemented with 10% (v/v) fetal bovine serum (ThermoFisher Scientific), 2 mM L-glutamine, 1% penicillin-streptomycin, 10 mM sodium pyruvate in 5% $CO_2$ in air at 37°C. Cells were seeded on to coverslips in 24-well plates to reach a density of ~60% at the time of transfection. Cells were transfected with rat TRPV1-encoding DNA 48 hr before use in ICC experiments using FuGENE Transfection Reagent (Promega) according to manufacturers' instructions. At 48 hr post-transfection, Affimers were incubated on live cells at a final concentration of 5 µg/ml in assay buffer (130 mM NaCl, 10 mM glucose, 5 mM KCl, 2 mM CaCl$_2$, 1.2 mM MgCl$_2$, 10 mM HEPES, pH 7.4) for 20 min. Cells were then washed with the same buffer three times for 5 min per wash before fixation with 4% PFA for 10 min. Fixed cells were washed three times in PBS and then permeabilised with 0.1% Triton X100 in PBS. Cells were blocked for 30 min in 1% BSA in PBS at room temperature. Mouse monoclonal anti-6X His antibody (Abcam: Ab18184) was incubated on cells for one hour at room temperature. Cells were then washed three times with PBS and incubated for one hour in the dark with anti-mouse 488 antibody (Life Technologies: A11001) for 1 hr. A further two washes in PBS-T, two washes in PBS and one wash in ddH$_2$0 was followed by mounting on to glass slides with ProLong Diamond Antifade Mountant with DAPI (ThermoFisher Scientific). The next day, samples were imaged using an EVOS FL imaging system (ThermoFisher Scientific).

For co-localisation staining with anti-TRPV1 antibody, the same protocol was followed with an additional step in which a further 1 hr incubation was conducted with anti-TRPV1 antibody (Abcam: ab10295) post-fixation. Goat anti-guinea pig 647 (Abcam: ab150187) was then applied for the detection of anti-TRPV1 antibody for one hour at room temperature in the dark.

## TRPV1 modulation assay

U2-OS cells were cultured using the above conditions. Cells were seeded in to T75cm flasks so that confluence would reach ~60% by the time of transfection. 24 hr post-transfection with TRPV1 DNA, cells were trypsinised in to black-walled 96 well-plates (Greiner Bio-One) at a confluence of ~100,000 cells per well and incubated for a further 24 hr under previously described cell culture conditions.

On the day of the modulation assay, cells were washed with assay buffer and loaded with 50 µL Fluo-4 AM (1 µM) (ThermoFisher Scientific: F14201) for 1 hr at 37°C. Cells were washed again with 200 µL assay buffer prior to the addition of Affimer at a concentration of 1 µM in 50 µL assay buffer. Following 30 min of incubation at room temperature, the increase in intracellular $Ca^{2+}$ was measured using a Flexstation 3 (Molecular Devices; Sunnyvale, CA, USA). Fluorescence was detected for 60 sec at 485 nm excitation and 525 nm emission, but the peak $Ca^{2+}$ response (approximately 5 sec after addition of the orthosteric TRPV1 agonist, capsaicin) was used for the subsequent determination of the agonist response. Initially, the effect of Affimers on $Ca^{2+}$ response was tested at a capsaicin $EC_{20}$ concentration whilst subsequent experiments tested a range of capsaicin concentrations in a so-called curve shift assay. Relative peak fluorescence units were normalised to the response observed in the absence of Affimer. Data analysis - ordinary one-way ANOVA with multiple comparisons was conducted for modulation assays against an $EC_{20}$ concentration of capsaicin. $P$ values less than 0.05 (*) 0.01 (**) and 0.001 (***) and 0.0001 (****) are indicated.

## Characterisation of binding to TNT and DNT analogues by competition ELISA

Immulon2 HB 96-well micro-titre plates (Nunc) were coated with TNBS-ovalbumin conjugate at a concentration of 10 µg ml$^{-1}$ in PBS (100 µl per well) and incubated overnight at 5°C. Each well was washed three times with PBS containing 0.05 % v/v Tween$^{20}$ (PBST) prior to blocking using 2% (w/v) skimmed milk powder (Marvel) in PBST (blocking buffer) by incubating for 60 min at room temperature. Each well was then washed and free TNBS-Ovalbumin, TNT or DNT analogue was diluted in blocking buffer and added to 6 replicate wells for each Affimer at a concentration of 100 µg/ml. Each hapten was then serially diluted down the plate to a final concentration of 0.78 µg/ml in blocking buffer. Subsequently, 50 µl of biotinylated Affimers TNT3, 4 and 9 (0.5 µg/ml) and TNT15 (1.0 µg/ml) in blocking buffer was added in replicate to each of the wells containing free hapten to allow for six technical replicate dilution curves of each free hapten molecule with each Affimer to be generated (technical replicate; the number of times the experiment was repeated within one experiment). For the negative control, the Affimer was substituted for blocking buffer. Each plate was incubated at room temperature for 1 hr, washed three times with PBST and Affimer that remained bound to the TNBS-Ovalbumin conjugate was detected using a high sensitivity streptavidin-HRP conjugate (ThermoFisher Scientific), diluted 1:2000 in blocking buffer. The presence of HRP was detected using hydrogen peroxide (Sigma) and ABTS substrate (Sigma) in substrate buffer (0.1 M Citric acid, 0.2 M $Na_2HPO_4$ at pH 4.37) with the response quantified based on readings at 414 nm in an automated plate reader (Anthos 2001, Anthos Labtec Instruments).

## Immuno- and Affinity-histochemistry for tenascin C

SW620 (mycoplasma tested - RRID:CVCL_0547) xenograft mice were sacrificed; tumours were harvested and embedded in paraffin wax. Xenograft tissue was then processed for tenascin C immuno-histochemistry as follows. Briefly, 4 µm paraffin sections were cut and collected on poly-lysine coated slides. Sections were dewaxed in xylene solutions and rehydrated in graded alcohol followed by distilled water. Antigen retrieval was performed by pressure-cooking in 0.01 M citric acid buffer, pH 6.0. Following antigen retrieval, tissue sections were washed in distilled water and endogenous peroxidase was blocked with Bloxall blocking reagent (10 min, SP-6000; Vector Laboratories Ltd, Peterborough, UK). After washing with TBS-T, endogenous Avidin/Biotin was blocked using Avidin/Biotin blocking kit (SP-2001 Vector Laboratories Ltd, Peterborough, UK). Tissue sections were washed in TBS-T and non-specific protein binding sites were blocked using 1 X casein (20 min, SP-5020; Vector Laboratories Ltd, Peterborough, UK) prepared in antibody diluent (Sigma, Poole, UK). Sections were then incubated overnight (4°C) in mouse monoclonal anti-TNC antibody (1:25, 4F10TT, IBL, USA) and bound antibody was detected using the mouse on mouse polymer IHC kit (ab127055, Abcam, Cambridge, UK) according to the manufacturer's instructions. Tissue sections

were counterstained with haematoxylin, dehydrated, cleared and mounted in DPX. Images were captured an Axioplan Zeiss microscope and AxioVision 4.8 software (Carl Zeiss Inc. Germany).

Tissue staining using the TNC Affimer was performed in a similar mannerhowever, after blocking non-specific protein binding sites with casein, sections were incubated in biotinylated TNC-Affimer overnight at 4°C. Sections were then washed in TBS-T and bound Affimer was visualized using Streptavidin/HRP (1:300, 30 min, SA-5004;, Vector Laboratories Ltd, Peterborough, UK) with 3,3′-diaminobenzidine (DAB) as substrate (SK41-05; ImmPACT DAB, Vector Laboratories Ltd, Peterborough, UK). Sections were counterstained and imaged as described previously.

## Immuno- and Affinity-histochemistry for VEGFR2

Wax embedded tissue sections of human pancreatic tissue were collected and processed for immunostaining in a manner similar to that described above. After dewaxing and rehydrating the tissue sections, antigen retrieval for the VEGFR-2 epitope was carried out using Tris EDTA buffer (pH 9.0). Endogenous peroxidase, Avidin/Biotin and protein were blocked as described and tissue sections were incubated overnight at 4°C in rabbit monoclonal anti VEGFR2 antibody (1:25, clone 55B11, Cell Signaling Technology, Danvers, USA) or VEGFR2 Affimers (1–11 μg/ml). Bound antibody was visualized using polyclonal goat anti-rabbit biotinylated antibody (1:200, 30 min, clone E0432, DAKO, UK) and Streptavidin/HRP with DAB as substrate. Affimers were visualized using Streptavidin/HRP with DAB as substrate. Section were counterstained and imaged as described previously (n = 3).

## Cell culture and transfection

MCF7 cells (purchased from ECACC, STR profiled and mycoplasma negative - RRID:CVCL_0031) were grown in RPMI medium supplemented with 10% (v/v) fetal bovine serum (ThermoFisher Scientific), 2 mM L-glutamine, 1% penicillin-streptomycin, 10 mM sodium pyruvate in 5% $CO_2$ in air at 37°C. Mouse fibroblasts, NIH-3T3, were cultured in Dulbecco's Modified Eagle's Medium (Sigma) with 10% FCS and 2 mM L-glutamine in a humidified atmosphere at 37°C in 5% $CO_2$. Affimers specific to the N-terminal SH2 domain of p85 were transfected into NIH-3T3 using TransIT 293 (Mirus, Madison, USA) according to the manufacturers' instructions.

Human umbilical vein endothelial cells (HUVECs) were isolated and cultured in endothelial cell growth medium (ECGM). Human umbilical cords used for isolation of primary endothelial cells were provided by written informed consent in accordance with ethical guidelines and under ethical approval (reference CA03/020) of the Leeds NHS Hospitals Local Ethics Committee (UK). HUVECs were seeded into 6-well plates and cultured (for at least 24 hr) in ECGM until ~80% confluent, washed twice in PBS and then starved in MCDB131 plus 0.2% (w/v) BSA for 2–3 hr. HUVECS were treated with 0, 50, 100 or 150 μg/ml Affimer for 30 min prior to stimulation with 25 ng/ml VEGF-A (Genentech Inc., San Francisco, USA) for 0, 5 or 15 min.

Chicken Embryonic Fibroblasts (CEF) were derived from 10 day embryos and maintained in E199 medium (Sigma) supplemented with 10% Tryptose phosphate broth (BD) and 5% foetal calf serum (Sigma), at 38.5°C. Virus infections were generated by lipofectamine transfection of BAC clones of either MDV-1 (RB1B strain), HVT or DEV (2085 strain). Briefly, 2 μg of BAC DNA was diluted into 100 μl of Opti-MEM reagent, and mixed with 10 μl lipofectamine (Invitrogen) diluted in 100 μl Opti-MEM (Invitrogen). After complex formation for 30 min a further 800 μl of Opti-MEM was added and the 1 ml sample transferred onto a well of a six well plate containing CEFs that had been rinsed twice with Opti-MEM. DNA complexes were left on CEFs for 6 hr in normal culture conditions after which 2 ml of growth medium was added per well and returned to the incubator. Once virus replication was established infected cells were passaged onto fresh CEFs as required to maintain the infection.

## Affinity-fluorescence and in-cell western analysis

For immunofluorescence and in-cell Western experiments primary CEFs were infected with either MDV-1 (strain RB1B), HVT, or DEV (strain 2085). In all cases virus was derived from BAC clones by transfection into CEFs, these viruses constitutively express a GFP marker under the control of the thymidine kinase promoter present within the BAC element. For in-cell Western studies, infected cells were seeded into 96 well plates and allowed to adhere overnight. Cells were then fixed with 4% paraformaldehyde in PBS, washed with PBS, permeabilsed with 0.1% TritonX100 in PBS and blocked with 0.5% BSA in PBS for 30 min at room temperature. Affimers were then added at 1.5 μg/

ml or a goat polyclonal anti-GFP antibody (Sicgen) at 1:2000 dilution in blocking buffer and incubated at room temperature for 1 hr. Samples were then extensively washed with PBSa before secondary antibody in blocking buffer was added. For in-cell Western, donkey anti goat 680 was used at 1:5000 dilution, with Streptavidin 800 conjugate at 1:5000 dilution (Licor), and incubated for 1 hr at room temperature, then extensively washed with PBS. In-cell Westerns were imaged using the Licor Odyssey system using both the 700 nm and 800 nm channels following the manufacturer's recommendations. Images were exported from the manufacturer's proprietary software and processed using Adobe Illustrator. For immunofluorescence the same approach was followed after seeding of cells to coverslips. Coverslips were incubated with Affimers as the only primary detection reagent, with subsequent labelling with streptavidin-568 conjugate (Invitrogen) at 1:1000 dilution. After washing off excess Streptavidin-568, cells were stained with DAPI, followed by three deionised water washes before mounting on a Vectashield mounting medium (Vector Laboratories). Immunofluorescence images were captured using a Leica SP5 system and manufacturer's software, from the 488 nm channel, 568 nm channel and the 405 nm channel using the 63 x objective and treated as for in-cell Western images.

## Determination of dissociation equilibrium constants

Amine coupling chips (sensor chip CM5, GE Healthcare) were primed in 0.1 M sodium acetate pH 5.6 and functionalized with EDC/NHS35 µl at 5 µL/min. Target protein (1 mg/mL) was immobilized to one flow cell (300–600 response units), at a flow rate of 5 µL/min and the flow cell was capped with 1 M ethanolamine-HCl (35 µL at 5 µL/min). The non-functionalised flow cell (acting as a blank) was treated with EDC/NHS (35 µL at 5 µL/min) and 1 M ethanolamine-HCl (35 µL at 5 µL/min. The system was primed in PBS supplemented with 0.1% TritonX 100. Five concentrations of Affimer (5–500 nM) were tested. Each concentration was flowed over both the functionalised and the non-functionalised flow cells at 40 µL/min and the association and dissociation rate constants $k_a$ and $k_d$, respectively were calculated using the Biacore software allowing determination of the equilibrium dissociation constant, $K_D$ as below.

$$d[AB]/-dt = k_a[A][B] \bullet d[AB]/dt = k_d[AB]$$
$$K_D = k_d/k_a$$

$K_D$ values were also determined using the Octet Red interferometer (Pall Fortebio) using streptavidin coated biosensors (AMC, 18–5019) as previously described (**Kumaraswamy and Tobias, 2015**). All experiments were carried out in HBS-EP buffer (10 mM HEPES (pH 7.4), 150 mM NaCl, 3 mM EDTA, 0.005% (v/v) Tween 20). $K_D$ values were determined by binding each biotinylated Affimer to a row of AMC biosensors at a constant concentration of 50 nM. Next a 2-fold dilution of unlabelled purified protein starting at 41 nM was bound to the Affimers. Raw offset values were plotted against concentration of purified HVT UL49 protein, determined by densitometry, and modelled to one site-specific binding equation using Graphpad Prism 6.

## Labelling of Affimers with rhodamine red

The C-terminal cysteine residues of Affimers TNC15C and GFP32C were labeled with Rhodamine Red $C_2$ maleimide (Thermo Fisher Scientific). Samples of Affimer (TNC15C or GFP32C) (80–200 µM) in elution buffer (50 mM NaH$_2$PO$_4$, 500 mM NaCl, 300 mM imidazole, 10% glycerol, pH 7.4) were dialysed (2 X with a dilution of 1000x) into labelling buffer (PBS containing 20% glycerol and 0.05% Tween-20; pH 7.4). The samples were then treated with TCEP in H$_2$O at 2.5 mM and Rhodamine Red $C_2$ maleimide (20 mM in DMSO; 5 equiv.) and rocked for 6 hr. Upon completion assessed by mass spectrometry, the reactions were quenched with $\beta$-mercaptoethanol (100 equiv.) and the mixture was spin concentrated (3 kDa cut-off). The concentrated mixture was passed through a buffer exchange column (PD-10, GE Healthcare), eluting 0.5 mL fractions with labelling buffer. Fractions containing protein were identified by BioRad colorimetric assay and pooled taking care not to include fractions containing free Rhodamine Red dye that elute later. The labelled Affimers were then concentrated to 250–300 µM in a spin concentrator (3 kDa cut-off). The concentrations were estimated by SDS-PAGE analysis against known amounts of BSA as standard. The identities of labelled Affimers were confirmed by mass spectrometry. Samples were used immediately or flash frozen in liquid nitrogen and stored at −80°C until required.

## In vivo work

All procedures were carried out in accordance with the Animals (Scientific Procedures) Act 1986 under project licence approval (PPL 70/7965). Ethical review and monitoring was undertaken by the Animal Welfare and Ethics Review Committee (AWERC) at the University of Leeds. Twenty-four 6–10 week old BALB/c nude female mice (originally obtained from Charles River, UK then maintained in-house) were injected subcutaneously in the right flank with $1 \times 10^7$ SW620 cells (obtained from ECACC and verified by single tandem repeat analysis). After 10–14 days of tumour growth, animals were randomised to receive either the tenascin C Affimer or a control GFP Affimer conjugated with Rhodamine Red $C_2$ maleimide, via tail vein injection (mean tumour volume tenascin C Affimer group $316.9 \pm 192.0$ mm$^3$ vs $360.8 \pm 216.6$ mm$^3$ in control GFP Affimer group; p=0.64). Approximately 300 µM labelled Affimer in 100 µL PBS with 20% glycerol and 0.05% Tween-20 was injected intravenously into each animal. Fluorescent images of harvested tissues (tumour, liver, kidney, spleen, heart, lung and brain) were captured ex vivo using IVIS Spectrum (excitation 570 nm, emission 620 nm; Perkin Elmer, USA) Fluorescence intensity (radiant efficiency in p/s/cm$^2$/sr/µW/cm$^2$) for each tissue was determined for a region of interest of defined unit area using Living Image software (v4.3.1, Perkin Elmer). Mean background fluorescence intensity was normalized to sham injected control tumours and organs.

## Super-resolution microscopy methods

### HER4: (a) Cell labelling for binding curves

Chinese Hamster Ovary (CHO) cells (STR profiled and mycoplasma tested) were seeded at a density of $0.75 \times 10^5$ cells dish on uncoated 35 mm no. 1.5 glass-bottomed dishes (MatTek Corporation, USA). Cells were transfected with HER4-CYT-eGFP the next day and were serum starved overnight the second day after transfection. MCF7 were seeded and grown for two days, before being starved overnight.

Cells were rinsed and cooled to 4°C for 10 min, then labelled with HER4 Affimer labelled with CF640R (Biotium) in concentrations ranging from 1 to 100 nM (CHO/HER4-CYT-eGFP) or from 10 to 500 nM (MCF7). Labelling was carried out at 4°C for 1 hr. Cells were rinsed and fixed with 3% para-formaldehyde plus 0.5% glutaraldehyde for 15 min at 4°C then 15 min at room temperature. Imaging was carried out in PBS using the standard confocal mode in a Leica TCS sp8 (see Data analysis of binding curves below).

### HER4: (b) Data analysis of binding curves

Fluorescence images acquired in photon counting mode were analysed using Fiji (ImageJ). Membrane signals were isolated using intensity threshold segmentation followed by dilation of the resulting binary mask. For each Affimer concentration, the pixel values of the segmented membrane areas from five imaged regions were combined to produce a histogram of pixel values. The mean and standard deviation of the pixel value distributions was plotted.

### HER4: (c) Cell labelling for super-resolution imaging

CHO cells ($1 \times 10^5$) were seeded in untreated 35 mm high-precision glass bottomed-dishes (MatTek Corporation, USA) and reverse transfected with HER4-CYT-eGFP at a ratio of 4 ul ViaFect (Promega) to 1 ug DNA.Two days later cells were labelled with 100 nM HER4 Affimer5 conjugated to Alexa 647 for 1 hr at 4°C. Cells were rinsed and fixed with 3% paraformaldehyde plus 0.5% glutaraldehyde for 15 min at 4°C then 15 min at room temperature before rinsing.

### HER4: (d) dSTORM imaging

The super-resolution dSTORM images were taken in a Zeiss Elyra PS.1 system. The fluorophore Alexa Fluor 647 was photo-switched using 642 nm and 405 nm excitation lasers simultaneously, with 100 mM dithiothreitol in PBS as the switching buffer. The power density of the 642 nm and 405 nm illumination on the sample plane were about 4.6 kW/cm$^2$ and 0.4 kW/cm$^2$. A 100× NA 1.46 oil immersion objective lens (Zeiss alpha Plan-Apochromat) and a multi-band dichroic filter (BP 420–480+ LP 650) were used in the imaging. The final fluorescent images were projected on an Andor iXon 897 EMCCD camera. Super-resolution dSTORM images were reconstructed in ZEISS ZEN

software. A 25 nm localisation precision is obtained according to the histogram of the calculated localization precision from each fluorophore.

## HER4: (e) Cell labelling for single-molecule tracking experiments

MCF7 cells were seeded at the density of $3 \times 10^5$ cells/dish onto 35 mm glass bottomed-dishes. Each dish contains a high-precision no. 1.5 ± 0.170 thickness glass bottom of 14 mm in diameter (MatTek Corporation, USA). The dishes were cleaned with Piranha solution and coated with 1% BSA, according to the protocol previously described (*Zanetti-Domingues et al., 2012*). Prior to imaging, cells were starved for 2 hr at 37°C in serum free medium supplemented with 25 mM HEPES. After starvation, cells were rinsed twice with serum free medium pre-heated at 37°C. Labelling with fluorescently labelled Affimer was carried out for 10 min at 37°C. HER4 Affimer 5 (HER4-5) was conjugated in-house with CF640R (Biotium) maleimide dye following manufacturers' instructions. Cells were rinsed twice with serum free medium pre-heated at 37°C and promptly imaged as described below.

## HER4: (f) Single-molecule acquisition

Single-molecule images were acquired using an Axiovert 200M microscope with a TIRF illuminator (Zeiss, UK) and incorporating a 100 x oil-immersion objective ($\alpha$-Plan-Fluar, NA = 1.45; Zeiss, UK) and an EMCCD (iXon X3; Andor, UK). Samples were illuminated with a 638 nm laser (100 mW, Vortran) fed into the microscope via a polarisation maintaining triple laser combiner (Oz Optics) . Alternatively, the 640 nm lines of a Vortran Combiner or of an Andor Revolution Laser Combiner were used. A wrap-around incubator (Pecon XL S1) was used to maintain a constant temperature of 37°C. The field of view of each channel was 80 × 30 µm. Data were acquired at 20 Hz for 30 s. Images were saved in HDF5 format for subsequent processing using custom-designed software. All Single-Molecule time series data were analysed using the multidimensional analysis software described previously (*Rolfe et al., 2011*).

## Tubulin 3D direct stochastic optical reconstruction microscopy (3D dSTORM)

### (a) Sample preparation

Affinity purified monoclonal rat anti-tubulin IgG2a antibody, clone YL1/2, (BioRad Antibodies) described previously (*Kilmartin et al., 1982*; *Wehland et al., 1983*) was directly labeled with Alexa Fluor 647. Briefly, 20 µg of antibody was incubated with PBS containing 120 mM $NaHCO_3$ and 0.4 µg carboxylic acid succinimidyl ester Alexa Fluor 647 (A37573, Life Technologies Inc.) for 30 min at room temperature. Unincorporated dye was removed by gel filtration using NAP-5 columns (17-0853-02, GE Healthcare) following the manufacturers protocol. Antibody:dye labelling ratios of approximately 1:1 were confirmed by measured absorbance in a spectrophotometer, with a final concentration of 0.2 µg/µl. The C-terminal cysteine of Affimer 32, raised against tubulin, was labelled immediately after purification with the maleimide derivative of Alexa Fluor 647 (Thermo Fisher Scientific). Briefly, 150 µl of immobilized TCEP disulphide-reducing gel was washed three times with PBS containing 1 mM EDTA before being resuspended in 4 µl PBS containing 50 mM EDTA. The gel was incubated with 150 µl Affimer prepared at 0.5 mg/ml in PBS for 1 hr at room temperature to reduce the cysteine ready for labelling. The reduced Affimer was centrifuged at 1000 rpm for 1 min to pellet the gel and 130 µl of supernatant containing the Affimer was mixed with 6 µl of a 2 mM Alexa-647 maleimide stock and incubated at room temperature for 2 hr. Unbound Alexa Fluor 647 was removed by passing the labelled Affimer through a Zeba Spin Desalting Column, 7K MWCO (Thermo Scientific) according to the manufacturer's instructions. The labelled Affimer was stored at 4°C at a final concentration of 0.5 mg/ml.

Coverslips (#1.5, 25 mm diameter; Scientific Laboratory Supplies, MIC3350) were cleaned as described previously (*Shroff et al., 2007*). HeLa cells were seeded at $2 \times 10^5$ cells per coverslip in 30 mm diameter culture dishes in DMEM (Gibco) supplemented with 10% FCS, 1% P/S and incubated at 37°C, 5% $CO_2$ for 24 hr. Cells were fixed in 2% paraformaldehyde (PFA) dissolved in PEM buffer (80 mM PIPES pH 6.8, 5 mM EGTA, 2 mM $MgCl_2$) supplemented with 0.1% glutaraldehyde for 20 min at room temperature prior to being processed for immunofluorescence staining. Cells were permeabilised with 0.5% Triton X-100 for 5 min, washed three times with PBS before blocking

with 5% BSA in PBS for 1 hr. Cells were incubated with either directly labelled anti-tubulin antibody (1:20) or labelled Affimer (1:2000) prepared in PBS supplemented with 0.25% BSA for 1 hr at room temperature. Coverslips were washed three times prior to imaging. Dilutions of antibody and Affimer were previously optimised to give the best performance in dSTORM. For standard confocal imaging smaller (13 mm diameter) coverslips were used, the dilution of the antibodies and Affimers was 1/500 and 1/100 respectively, and a secondary antibody (labelled with Alexa Fluor 488, Life Technologies) to the rat anti-tubulin antibody was used at 1:400. The fixation and staining procedure was the same, and coverslips were mounted in Pro-long Antifade before imaging on a Zeiss LSM880 confocal, equipped with an Airyscan.

For dSTORM imaging, coverslips were mounted into the imaging chambers and placed on the microscope stage. Coverslips were then incubated with 0.01% poly-L-lysine (Sigma-Aldrich, P4707) for 10 min, followed by a suspension of 150 nm gold nanoparticles (Sigma-Aldrich, 742058, 1:10 in PBS), serving as stable reference emitters for calibration images and drift-correction. dSTORM data was acquired in the presence of fluorescence quenching buffer (*Heilemann et al., 2009*, *2008*) (PBS (pH 8), 5 mg/ml glucose, 114 mM $\beta$-mercaptoethanol, 0.5 mg/ml glucose oxidase and 40 µg/ml catalase).

## (b) Image acquisition and reconstruction

The 3D dSTORM system was based on an inverted microscope (Olympus, IX81) as previously described (*Lambacher et al., 2016*), with a 60x, 1.2 NA, water-immersion objective lens (Olympus, UPLSAPO60XW), and a cylindrical lens with $f$ = 150 mm (Thorlabs, LJ1629RM-A) for generating astigmatism. Lasers at 642 nm and 405 nm (Omicron, LuxX) provided widefield excitation and photo-activation of Alexa Fluor 647, together with a 2x beam expander before the rear illumination port of the microscope. Images were captured by a back-thinned, electron-multiplying CCD camera (EMCCD), cooled to −80°C (Andor Technology, iXON Ultra, model DU-897U-CSO-#BV), using published scripts (*York et al., 2011*) called from the camera interface (Andor Technology, SOLIS).

The acquisition workflow has been described previously (*York et al., 2011*) (see https://github.com/AndrewGYork/palm3d for further details), including capture of calibration images of a gold nanoparticle in steps of 50 nm in $z$ over a 4 µm range. The fluorescent dye labels in the sample were excited using a 642 nm laser emitting 100 mW, until a sufficient number were quenched (*Heilemann et al., 2009*, *2008*) for spatially nearby emission events to be separated in time, following which we began data collection (exposure time 50 ms, EMCCD gain 150). When emission events became sparse (after some 10,000 s of frames), labels were stochastically re-activated (*van de Linde et al., 2011*) using a 405 nm laser, with increasing power from 2 to 20 µW. Data collection finished when the number of emission events per frame became negligible. Emission events lasting for more than one frame were linked into averaged localisations, which were finally binned into a histogram for display, accounting for distortion by the cylindrical lens.

## Acknowledgements

We would like to acknowledge L McVeigh and A Charalambous for technical assistance with the in vivo work, and D J Williamson and G M Burslem for technical assistance with protein labelling. For funding we thank the Biomedical Health Centre at the University of Leeds, Leeds Teaching Hospital Charitable Foundation, Leeds Teaching Hospital NHS Trust, the University of Leeds Wellcome Trust ISSF fellowship award, The British Heart Foundation (NH/12/1/29382), Yorkshire Cancer Research (L362), and the Medical Research Council (MR/K015613/1).

## Additional information

### Competing interests

HK, GWP, AS, PKF, MJ: Works for Avacta Life Sciences who have commercially exploited this technology. RSB: Owns personal shares in Avacta Life Sciences. MJM: Co-inventor of the Adhiron/Affimer technology and owns personal shares in Avacta Life Sciences. The Adhiron patent (patent application number PCT/GB2014/050435) is owned by the University of Leeds and licensed to Avacta Ltd. DCT: Co-inventor of the Adhiron/Affimer technology. The Adhiron patent (patent

application number PCT/GB2014/050435) is owned by the University of Leeds and licensed to Avacta Ltd. The other authors declare that no competing interests exist.

## Funding

| Funder | Grant reference number | Author |
|---|---|---|
| Medical Research Council | | Alistair Curd<br>Michelle Peckham |
| Wellcome | | Ruth E Hughes |
| British Heart Foundation | NH/12/1/29832 | Iain Manfield<br>Azhar Maqbool<br>Mark Drinkhill<br>Robin S Bon |
| Yorkshire Cancer Research | L362 | Margaret Knowles |

The funders had no role in study design, data collection and interpretation, or the decision to submit the work for publication.

## Author contributions

CT, Conceptualization, Data curation, Formal analysis, Supervision, Validation, Investigation, Methodology, Writing—original draft, Writing—review and editing; RB, Conceptualization, Investigation, Methodology, Writing—original draft; SJH, HK, Data curation, Formal analysis, Supervision, Investigation, Methodology, Writing—original draft; GS, IW, RR, DA, APER, SRN, LCZ-D, AAT, FE, VP, SAG, Data curation, Formal analysis, Investigation, Methodology, Writing—original draft; AB, Data curation, Investigation; AC, Data curation, Supervision, Methodology, Writing—original draft, Writing—review and editing; REH, Data curation, Methodology, Writing—original draft, Writing—review and editing; HM, NI, Data curation, Investigation, Methodology; YS, TPP, TT, Data curation, Formal analysis, Investigation, Methodology; NG, Formal analysis, Investigation, Methodology; GWP, AS, Resources, Formal analysis, Supervision, Methodology; LPC, SB, Formal analysis, Supervision, Investigation; IM, Data curation, Supervision, Funding acquisition; MK, Supervision, Funding acquisition, Methodology; AM, Data curation, Formal analysis, Funding acquisition, Investigation, Methodology, Writing—original draft; RKP, Supervision, Methodology, Writing—original draft; MD, Formal analysis, Supervision, Funding acquisition, Writing—original draft; RSB, Data curation, Formal analysis, Supervision, Funding acquisition, Methodology, Writing—original draft; MM-F, Supervision, Methodology, Writing—original draft, Writing—review and editing; RJO, Supervision, Investigation, Methodology, Writing—original draft; JEN, Data curation, Methodology, Writing—original draft; MEW, Resources, Methodology, Writing—original draft, Writing—review and editing; MH, Conceptualization, Supervision, Investigation; JDL, Supervision, Investigation, Methodology; SP, Conceptualization, Supervision, Investigation, Methodology; MP, Data curation, Formal analysis, Supervision, Methodology, Writing—original draft, Writing—review and editing; PKF, Supervision, Writing—original draft; MJ, Conceptualization, Resources, Formal analysis, Supervision, Methodology; MJM, Conceptualization, Formal analysis, Supervision, Funding acquisition, Methodology, Writing—original draft, Writing—review and editing; DCT, Conceptualization, Resources, Data curation, Formal analysis, Supervision, Funding acquisition, Investigation, Methodology, Writing—original draft, Project administration, Writing—review and editing

## Author ORCIDs

Anna A Tang, http://orcid.org/0000-0002-5216-5080
Nicola Ingram, http://orcid.org/0000-0001-5274-8502
Sreenivasan Ponnambalam, http://orcid.org/0000-0002-4452-7619
Michelle Peckham, http://orcid.org/0000-0002-3754-2028
Michael J McPherson, http://orcid.org/0000-0002-0719-6427
Darren Charles Tomlinson, http://orcid.org/0000-0003-4134-7484

## Ethics

Human subjects: Human umbilical vein endothelial cells (HUVECs) were isolated and cultured in endothelial cell growth medium (ECGM). Human umbilical cords used for isolation of primary endothelial cells were provided by written informed consent in accordance with ethical guidelines and

under ethical approval (reference CA03/020) of the Leeds NHS Hospitals Local Ethics Committee (UK).

Animal experimentation: All procedures were carried out in accordance with the Animals (Scientific Procedures) Act 1986 under project licence approval (PPL 70/7965). Ethical review and monitoring was undertaken by the Animal Welfare and Ethics Review Committee (AWERC) at the University of Leeds.

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
