## [Decision Letter]

Thank you for submitting your article "Affimer: versatile and renewable affinity reagents" for consideration by *eLife*. Your article has been favorably evaluated by Anna Akhmanova (Senior Editor) and three reviewers, one of whom is a member of our Board of Reviewing Editors. The following individuals involved in review of your submission have agreed to reveal their identity: Peter Kristensen (Reviewer #2); Savvas Savvides (Reviewer #3).

The reviewers have discussed the reviews with one another and the Reviewing Editor has drafted this decision to help you prepare a revised submission.

In this manuscript, Tiede et al. report on the successful selection of Affimers from a phage display library. Selections were done against various antigens, including a hapten, and the several downstream applications were exemplified like fluorescent labeling, expression in cells, histo-chemistry or in vivo injection. Altogether, the Affimers (also known as Adhiron) represent an attractive option when developing specific binders and this manuscript illustrate the power of this approach. The experiments are convincing and well done.

However, although the Affimer technology is impressive and would probably interest a large number of *eLife* readers, the way the manuscript is written is strongly biased. Whatever the quality of this novel scaffold is, it is not the only non-Ig scaffold developed to create orthogonal binders. There are also a lot of over-statements concerning the limitations of classical and recombinant antibodies or antibody fragments. In addition, Affimers were published before under the name of Adhiron and while the current manuscript gives an impressive vision of successful selection, the paper would strongly benefit from showing something new that was not achieved in the past by this lab.

To make the paper acceptable for *eLife*, the authors should extremely thoroughly revise the whole manuscript in order to carefully put Affimers in the proper historical and current context of competing technologies so that readers may get an exact idea of what this sort of approach brings. The authors should also attempt, if possible, to push one of their stories a bit further to reach goals unmatched before by the Affimers.

Essential revisions:

1) The manuscript systematically presents the Affimer approach as being the only non-antibody scaffold of any value (DARPins, FN-based or other similar tools are not even mentioned) and the only compact and robust scaffold usable in cells (no mention of single domain or nanobodies). In the current form, the manuscript reads more like an advertisement than a proper scientific paper. It is essential that a well-balanced overview and discussion of different approaches is provided. Please note that cosmetic changes (e.g. references to reviews on the subject) will not suffice.

2) The development of this scaffold as well as the library used here have been reported before (PEDS, 2014; Tiede et al., 2014) and the current manuscript is simply extending what was published before by this group (and as stated by the authors more than 350 selections were already carried out). Although it is not that clear, it appears that the anti-Grb2 screen is already presented in the 2014 paper. The novelty and the advantages of Affimer approach, compared to, e.g. nanobodies, should be illustrated and discussed more clearly. Ideally, a functional or therapeutic proof-of-concept application of Affimers should be presented.

Title: It is not clear what "renewable" means in this context. Please either make this more clear in the manuscript, or change the title.

---

## [Author Response]

Essential revisions:

1) The manuscript systematically presents the Affimer approach as being the only non-antibody scaffold of any value (DARPins, FN-based or other similar tools are not even mentioned) and the only compact and robust scaffold usable in cells (no mention of single domain or nanobodies). In the current form, the manuscript reads more like an advertisement than a proper scientific paper. It is essential that a well-balanced overview and discussion of different approaches is provided. Please note that cosmetic changes (e.g. references to reviews on the subject) will not suffice.

Clearly the purpose of this paper is to describe a range of applications to which Affimers have been applied, however, we acknowledge that there was too little recognition of other complementary technologies. We have now altered the manuscript throughout to reference other approaches and to provide a more balanced view and indicating, where appropriate, how these have been used in similar applications, which also justifies some of the proteins targeted using Affimers.

2) The development of this scaffold as well as the library used here have been reported before (PEDS, 2014; Tiede et al., 2014) and the current manuscript is simply extending what was published before by this group (and as stated by the authors more than 350 selections were already carried out). Although it is not that clear, it appears that the anti-Grb2 screen is already presented in the 2014 paper. The novelty and the advantages of Affimer approach, compared to, e.g. nanobodies, should be illustrated and discussed more clearly. Ideally, a functional or therapeutic proof-of-concept application of Affimers should be presented.

The anti-Grb2 work reported in this manuscript was a new screen that is distinct from that reported in the initial descriptive publication – this has been addressed in the text.

We have now added a new figure demonstrating modulation of ion channels as a functional and therapeutic proof-of-concept application of Affimers.

Title: It is not clear what "renewable" means in this context. Please either make this more clear in the manuscript, or change the title.

The title has been changed as requested.